# Barrier Frank-Wolfe for Marginal Inference

**Rahul G. Krishnan**
Courant Institute
New York University

**Simon Lacoste-Julien**
INRIA - Sierra Project-Team
École Normale Supérieure, Paris

**David Sontag**
Courant Institute
New York University

## Abstract

We introduce a globally-convergent algorithm for optimizing the tree-reweighted (TRW) variational objective over the marginal polytope. The algorithm is based on the conditional gradient method (Frank-Wolfe) and moves pseudomarginals within the marginal polytope through repeated maximum a posteriori (MAP) calls. This modular structure enables us to leverage black-box MAP solvers (both exact and approximate) for variational inference, and obtains more accurate results than tree-reweighted algorithms that optimize over the local consistency relaxation. Theoretically, we bound the sub-optimality for the proposed algorithm despite the TRW objective having unbounded gradients at the boundary of the marginal polytope. Empirically, we demonstrate the increased quality of results found by tightening the relaxation over the marginal polytope as well as the spanning tree polytope on synthetic and real-world instances.

## 1 Introduction

Markov random fields (MRFs) are used in many areas of computer science such as vision and speech. Inference in these undirected graphical models is generally intractable. Our work focuses on performing approximate marginal inference by optimizing the Tree Re-Weighted (TRW) objective (Wainwright et al., 2005). The TRW objective is concave, is exact for tree-structured MRFs, and provides an upper bound on the log-partition function.

Fast combinatorial solvers for the TRW objective exist, including Tree-Reweighted Belief Propagation (TRBP) (Wainwright et al., 2005), convergent message-passing based on geometric programming (Globerson and Jaakkola, 2007), and dual decomposition (Jancsary and Matz, 2011). These methods optimize over the set of pairwise consistency constraints, also called the local polytope. Sontag and Jaakkola (2007) showed that significantly better results could be obtained by optimizing over tighter relaxations of the marginal polytope. However, deriving a message-passing algorithm for the TRW objective over tighter relaxations of the marginal polytope is challenging. Instead, Sontag and Jaakkola (2007) use the conditional gradient method (also called Frank-Wolfe) and off-the-shelf linear programming solvers to optimize TRW over the cycle consistency relaxation. Rather than optimizing over the cycle relaxation, Belanger et al. (2013) optimize the TRW objective over the exact marginal polytope. Then, using Frank-Wolfe, the linear minimization performed in the inner loop can be shown to correspond to MAP inference.

The Frank-Wolfe optimization algorithm has seen increasing use in machine learning, thanks in part to its efficient handling of complex constraint sets appearing with structured data (Jaggi, 2013; Lacoste-Julien and Jaggi, 2015). However, applying Frank-Wolfe to variational inference presents challenges that were never resolved in previous work. First, the linear minimization performed in the inner loop is computationally expensive, either requiring repeatedly solving a large linear program, as in Sontag and Jaakkola (2007), or performing MAP inference, as in Belanger et al. (2013). Second, the TRW objective involves entropy terms whose gradients go to infinity near the boundary of the feasible set, therefore existing convergence guarantees for Frank-Wolfe do not apply. Third, variational inference using TRW involves both an outer and inner loop of Frank-Wolfe, where the outer loop optimizes the edge appearance probabilities in the TRW entropy bound to tighten it.

Neither Sontag and Jaakkola (2007) nor Belanger et al. (2013) explore the effect of optimizing over the edge appearance probabilities.

Although MAP inference is in general NP hard (Shimony, 1994), it is often possible to find exact solutions to large real-world instances within reasonable running times (Sontag et al., 2008; Allouche et al., 2010; Kappes et al., 2013). Moreover, as we show in our experiments, even approximate MAP solvers can be successfully used within our variational inference algorithm. As MAP solvers improve in their runtime and performance, their iterative use could become feasible and as a byproduct enable more efficient and accurate marginal inference. Our work provides a fast deterministic alternative to recently proposed Perturb-and-MAP algorithms (Papandreou and Yuille, 2011; Hazan and Jaakkola, 2012; Ermon et al., 2013).

**Contributions.** This paper makes several theoretical and practical innovations. We propose a modification to the Frank-Wolfe algorithm that optimizes over adaptively chosen contractions of the domain and prove its rate of convergence for functions whose gradients can be unbounded at the boundary. Our algorithm does not require a different oracle than standard Frank-Wolfe and could be useful for other convex optimization problems where the gradient is ill-behaved at the boundary.

We instantiate the algorithm for approximate marginal inference over the marginal polytope with the TRW objective. With an exact MAP oracle, we obtain the first provably convergent algorithm for the optimization of the TRW objective over the marginal polytope, which had remained an open problem to the best of our knowledge. Traditional proof techniques of convergence for first order methods fail as the gradient of the TRW objective is not Lipschitz continuous.

We develop several heuristics to make the algorithm practical: a fully-corrective variant of Frank-Wolfe that reuses previously found integer assignments thereby reducing the need for new (approximate) MAP calls, the use of local search between MAP calls, and significant re-use of computations between subsequent steps of optimizing over the spanning tree polytope. We perform an extensive experimental evaluation on both synthetic and real-world inference tasks.

## 2 Background

**Markov Random Fields**: MRFs are undirected probabilistic graphical models where the probability distribution factorizes over cliques in the graph. We consider marginal inference on pairwise MRFs with $N$ random variables $X_1, X_2, \ldots, X_N$ where each variable takes discrete states $x_i \in \text{VAL}_i$. Let $G = (V, E)$ be the Markov network with an undirected edge $\{i, j\} \in E$ for every two variables $X_i$ and $X_j$ that are connected together. Let $\mathcal{N}(i)$ refer to the set of neighbors of variable $X_i$. We organize the edge log-potentials $\theta_{ij}(x_i, x_j)$ for all possible values of $x_i \in \text{VAL}_i$, $x_j \in \text{VAL}_j$ in the vector $\boldsymbol{\theta}_{ij}$, and similarly for the node log-potential vector $\boldsymbol{\theta}_i$. We regroup these in the overall vector $\vec{\boldsymbol{\theta}}$. We introduce a similar grouping for the marginal vector $\vec{\boldsymbol{\mu}}$: for example, $\mu_i(x_i)$ gives the coordinate of the marginal vector corresponding to the assignment $x_i$ to variable $X_i$.

**Tree Re-weighted Objective** (Wainwright et al., 2005): Let $Z(\vec{\boldsymbol{\theta}})$ be the partition function for the MRF and $\mathcal{M}$ be the set of all valid marginal vectors (the marginal polytope). The maximization of the TRW objective gives the following upper bound on the log partition function:

$$\log Z(\vec{\boldsymbol{\theta}}) \leq \min_{\boldsymbol{\rho} \in \mathbb{T}} \max_{\vec{\boldsymbol{\mu}} \in \mathcal{M}} \underbrace{\langle \vec{\boldsymbol{\theta}}, \vec{\boldsymbol{\mu}} \rangle + H(\vec{\boldsymbol{\mu}}; \boldsymbol{\rho})}_{\text{TRW}(\vec{\boldsymbol{\mu}}; \vec{\boldsymbol{\theta}}, \boldsymbol{\rho})}, \tag{1}$$

where the TRW entropy is:

$$H(\vec{\boldsymbol{\mu}}; \boldsymbol{\rho}) := \sum_{i \in V} (1 - \sum_{j \in \mathcal{N}(i)} \rho_{ij}) H(\boldsymbol{\mu}_i) + \sum_{(ij) \in E} \rho_{ij} H(\boldsymbol{\mu}_{ij}), \quad H(\boldsymbol{\mu}_i) := -\sum_{x_i} \mu_i(x_i) \log \mu_i(x_i). \tag{2}$$

$\mathbb{T}$ is the spanning tree polytope, the convex hull of edge indicator vectors of all possible spanning trees of the graph. Elements of $\boldsymbol{\rho} \in \mathbb{T}$ specify the probability of an edge being present under a specific distribution over spanning trees. $\mathcal{M}$ is difficult to optimize over, and most TRW algorithms optimize over a relaxation called the local consistency polytope $\mathbb{L} \supseteq \mathcal{M}$:

$$\mathbb{L} := \left\{ \vec{\boldsymbol{\mu}} \geq \mathbf{0}, \ \sum_{x_i} \mu_i(x_i) = 1 \ \forall i \in V, \ \sum_{x_i} \mu_{ij}(x_i, x_j) = \mu_j(x_j), \sum_{x_j} \mu_{ij}(x_i, x_j) = \mu_i(x_i) \ \forall \{i, j\} \in E \right\}.$$

The TRW objective $\text{TRW}(\vec{\boldsymbol{\mu}}; \vec{\boldsymbol{\theta}}, \boldsymbol{\rho})$ is a globally concave function of $\vec{\boldsymbol{\mu}}$ over $\mathbb{L}$, assuming that $\boldsymbol{\rho}$ is obtained from a valid distribution over spanning trees of the graph (i.e. $\boldsymbol{\rho} \in \mathbb{T}$).

**Frank-Wolfe (FW) Algorithm:** In recent years, the Frank-Wolfe (aka conditional gradient) algorithm has gained popularity in machine learning (Jaggi, 2013) for the optimization of convex

functions over compact domains (denoted $\mathcal{D}$). The algorithm is used to solve $\min_{\boldsymbol{x} \in \mathcal{D}} f(\boldsymbol{x})$ by iteratively finding a good descent vertex by solving the linear subproblem:

$$\boldsymbol{s}^{(k)} = \arg\min_{\boldsymbol{s} \in \mathcal{D}} \langle \nabla f(\boldsymbol{x}^{(k)}), \boldsymbol{s} \rangle \qquad \text{(FW oracle)}, \tag{3}$$

and then taking a convex step towards this vertex: $\boldsymbol{x}^{(k+1)} = (1 - \gamma)\boldsymbol{x}^{(k)} + \gamma \boldsymbol{s}^{(k)}$ for a suitably chosen step-size $\gamma \in [0, 1]$. The algorithm remains within the feasible set (is projection free), is invariant to affine transformations of the domain, and can be implemented in a memory efficient manner. Moreover, the FW gap $g(\boldsymbol{x}^{(k)}) := \langle -\nabla f(\boldsymbol{x}^{(k)}), \boldsymbol{s}^{(k)} - \boldsymbol{x}^{(k)} \rangle$ provides an upper bound on the suboptimality of the iterate $\boldsymbol{x}^{(k)}$. The primal convergence of the Frank-Wolfe algorithm is given by Thm. 1 in Jaggi (2013), restated here for convenience: for $k \geq 1$, the iterates $\boldsymbol{x}^{(k)}$ satisfy:

$$f(\boldsymbol{x}^{(k)}) - f(\boldsymbol{x}^*) \leq \frac{2C_f}{k+2}, \tag{4}$$

where $C_f$ is called the "curvature constant". Under the assumption that $\nabla f$ is $L$-Lipschitz continuous[1] on $\mathcal{D}$, we can bound it as $C_f \leq L \operatorname{diam}_{||\cdot||}(\mathcal{D})^2$.

**Marginal Inference with Frank-Wolfe:** To optimize $\max_{\vec{\boldsymbol{\mu}} \in \mathcal{M}} \operatorname{TRW}(\vec{\boldsymbol{\mu}}; \vec{\boldsymbol{\theta}}, \boldsymbol{\rho})$ with Frank-Wolfe, the linear subproblem (3) becomes $\arg\max_{\vec{\boldsymbol{\mu}} \in \mathcal{M}} \langle \tilde{\boldsymbol{\theta}}, \vec{\boldsymbol{\mu}} \rangle$, where the perturbed potentials $\tilde{\boldsymbol{\theta}}$ correspond to the gradient of $\operatorname{TRW}(\vec{\boldsymbol{\mu}}; \vec{\boldsymbol{\theta}}, \boldsymbol{\rho})$ with respect to $\vec{\boldsymbol{\mu}}$. Elements of $\tilde{\boldsymbol{\theta}}$ are of the form $\theta_c(x_c) + K_c(1 + \log \mu_c(x_c))$, evaluated at the pseudomarginals' current location in $\mathcal{M}$, where $K_c$ is the coefficient of the entropy for the node/edge term in (2). The FW linear subproblem here is thus equivalent to performing MAP inference in a graphical model with potentials $\tilde{\boldsymbol{\theta}}$ (Belanger et al., 2013), as the vertices of the marginal polytope are in 1-1 correspondence with valid joint assignments to the random variables of the MRF, and the solution of a linear program is always achieved at a vertex of the polytope. The TRW objective does not have a Lipschitz continuous gradient over $\mathcal{M}$, and so standard convergence proofs for Frank-Wolfe do not hold.

## 3 Optimizing over Contractions of the Marginal Polytope

**Motivation**: We wish to (1) use the fewest possible MAP calls, and (2) avoid regions near the boundary where the unbounded curvature of the function slows down convergence. A viable option to address (1) is through the use of *correction steps*, where after a Frank-Wolfe step, one optimizes over the polytope defined by previously visited vertices of $\mathcal{M}$ (called the fully-corrective Frank-Wolfe (FCFW) algorithm and proven to be linearly convergence for strongly convex objectives (Lacoste-Julien and Jaggi, 2015)). This does not require additional MAP calls. However, we found (see Sec. 5) that when optimizing the TRW objective over $\mathcal{M}$, performing correction steps can surprisingly *hurt* performance. This leaves us in a dilemma: correction steps enable decreasing the objective without additional MAP calls, but they can also slow global progress since iterates after correction sometimes lie close to the boundary of the polytope (where the FW directions become less informative). In a manner akin to barrier methods and to Garber and Hazan (2013)'s local linear oracle, our proposed solution maintains the iterates within a contraction of the polytope. This gives us most of the mileage obtained from performing the correction steps *without* suffering the consequences of venturing too close to the boundary of the polytope. We prove a global convergence rate for the iterates with respect to the true solution over the full polytope.

We describe convergent algorithms to optimize $\operatorname{TRW}(\vec{\boldsymbol{\mu}}; \vec{\boldsymbol{\theta}}, \boldsymbol{\rho})$ for $\vec{\boldsymbol{\mu}} \in \mathcal{M}$. The approach we adopt to deal with the issue of unbounded gradients at the boundary is to perform Frank-Wolfe within a contraction of the marginal polytope given by $\mathcal{M}_\delta$ for $\delta \in [0, 1]$, with either a fixed $\delta$ or an adaptive $\delta$.

**Definition 3.1** (Contraction polytope). $\mathcal{M}_\delta := (1 - \delta)\mathcal{M} + \delta \boldsymbol{u}_0$, *where* $\boldsymbol{u}_0 \in \mathcal{M}$ *is the vector representing the uniform distribution.*

Marginal vectors that lie within $\mathcal{M}_\delta$ are bounded away from zero as all the components of $\boldsymbol{u}_0$ are strictly positive. Denoting $\mathcal{V}^{(\delta)}$ as the set of vertices of $\mathcal{M}_\delta$, $\mathcal{V}$ as the set of vertices of $\mathcal{M}$ and $f(\vec{\boldsymbol{\mu}}) := -\operatorname{TRW}(\vec{\boldsymbol{\mu}}; \vec{\boldsymbol{\theta}}, \boldsymbol{\rho})$, the key insight that enables our novel approach is that:

$$\underbrace{\arg\min_{\boldsymbol{v}^{(\delta)} \in \mathcal{V}^{(\delta)}} \left\langle \nabla f, \boldsymbol{v}^{(\delta)} \right\rangle}_{\textit{(Linear Minimization over } \mathcal{M}_\delta\textit{)}} \equiv \arg\min_{\boldsymbol{v} \in \mathcal{V}} \underbrace{\langle \nabla f, (1 - \delta)\boldsymbol{v} + \delta \boldsymbol{u}_0 \rangle}_{\textit{(Definition of } \boldsymbol{v}^{(\delta)}\textit{)}} \equiv \underbrace{(1 - \delta) \arg\min_{\boldsymbol{v} \in \mathcal{V}} \langle \nabla f, \boldsymbol{v} \rangle + \delta \boldsymbol{u}_0.}_{\textit{(Run MAP solver and shift vertex)}}$$

**Algorithm 1:** Updates to $\delta$ after a MAP call (Adaptive $\delta$ variant)

---

1: At iteration $k$. Assuming $\boldsymbol{x}^{(k)}, \boldsymbol{u}_0, \delta^{(k-1)}, f$ are defined and $\boldsymbol{s}^{(k)}$ has been computed
2: Compute $g(\boldsymbol{x}^{(k)}) = \langle -\nabla f(\boldsymbol{x}^{(k)}), \boldsymbol{s}^{(k)} - \boldsymbol{x}^{(k)} \rangle$    *(Compute FW gap)*
3: Compute $g_u(\boldsymbol{x}^{(k)}) = \langle -\nabla f(\boldsymbol{x}^{(k)}), \boldsymbol{u}_0 - \boldsymbol{x}^{(k)} \rangle$    *(Compute "uniform gap")*
4: **if** $g_u(\boldsymbol{x}^{(k)}) < 0$ **then**
5:    Let $\tilde{\delta} = \frac{g(\boldsymbol{x}^{(k)})}{-4g_u(\boldsymbol{x}^{(k)})}$    *(Compute new proposal for $\delta$)*
6:    **if** $\tilde{\delta} < \delta^{(k-1)}$ **then**
7:       $\delta^{(k)} = \min\left(\tilde{\delta}, \frac{\delta^{(k-1)}}{2}\right)$    *(Shrink by at least a factor of two if proposal is smaller)*
8:    **end if**
9: **end if**   *(and set $\delta^{(k)} = \delta^{(k-1)}$ if it was not updated)*

---

Therefore, to solve the FW subproblem (3) over $\mathcal{M}_\delta$, we can run as usual a MAP solver and simply shift the resulting vertex of $\mathcal{M}$ towards $\boldsymbol{u}_0$ to obtain a vertex of $\mathcal{M}_\delta$. Our solution to optimize over restrictions of the polytope is more broadly applicable to the optimization problem defined below, with $f$ satisfying Prop. 3.3 (satisfied by the TRW objective) in order to get convergence rates.

**Problem 3.2.** *Solve* $\min_{\boldsymbol{x} \in \mathcal{D}} f(\boldsymbol{x})$ *where* $\mathcal{D}$ *is a compact convex set and* $f$ *is convex and continuously differentiable on the relative interior of* $\mathcal{D}$.

**Property 3.3.** *(Controlled growth of Lipschitz constant over $\mathcal{D}_\delta$).* *We define* $\mathcal{D}_\delta := (1 - \delta)\mathcal{D} + \delta\boldsymbol{u}_0$ *for a fixed* $\boldsymbol{u}_0$ *in the relative interior of* $\mathcal{D}$. *We suppose that there exists a fixed* $p \geq 0$ *and* $L$ *such that for any* $\delta > 0$, $\nabla f(\boldsymbol{x})$ *has a bounded Lipschitz constant* $L_\delta \leq L\delta^{-p}$ $\forall \boldsymbol{x} \in \mathcal{D}_\delta$.

**Fixed $\delta$:** The first algorithm fixes a value for $\delta$ a-priori and performs the optimization over $\mathcal{D}_\delta$. The following theorem bounds the sub-optimality of the iterates with respect to the optimum over $\mathcal{D}$.

**Theorem 3.4** (Suboptimality bound for fixed-$\delta$ algorithm). *Let* $f$ *satisfy the properties in Prob. 3.2 and Prop. 3.3, and suppose further that* $f$ *is finite on the boundary of* $\mathcal{D}$. *Then the use of Frank-Wolfe for* $\min_{\boldsymbol{x} \in \mathcal{D}_\delta} f(\boldsymbol{x})$ *realizes a sub-optimality over* $\mathcal{D}$ *bounded as:*

$$f(\boldsymbol{x}^{(k)}) - f(\boldsymbol{x}^*) \leq \frac{2C_\delta}{(k+2)} + \omega\left(\delta \operatorname{diam}(\mathcal{D})\right),$$

*where* $\boldsymbol{x}^*$ *is the optimal solution in* $\mathcal{D}$, $C_\delta \leq L_\delta \operatorname{diam}_{||.||}(\mathcal{D}_\delta)^2$, *and* $\omega$ *is the modulus of continuity function of the (uniformly) continuous* $f$ *(in particular,* $\omega(\delta) \downarrow 0$ *as* $\delta \downarrow 0$).

The full proof is given in App. C. The first term of the bound comes from the standard Frank-Wolfe convergence analysis of the sub-optimality of $\boldsymbol{x}^{(k)}$ relative to $\boldsymbol{x}^*_{(\delta)}$, the optimum over $\mathcal{D}_\delta$, as in (4) and using Prop. 3.3. The second term arises by bounding $f(\boldsymbol{x}^*_{(\delta)}) - f(\boldsymbol{x}^*) \leq f(\tilde{\boldsymbol{x}}) - f(\boldsymbol{x}^*)$ with a cleverly chosen $\tilde{\boldsymbol{x}} \in \mathcal{D}_\delta$ (as $\boldsymbol{x}^*_{(\delta)}$ is optimal in $\mathcal{D}_\delta$). We pick $\tilde{\boldsymbol{x}} := (1 - \delta)\boldsymbol{x}^* + \delta\boldsymbol{u}_0$ and note that $\|\tilde{\boldsymbol{x}} - \boldsymbol{x}^*\| \leq \delta \operatorname{diam}(\mathcal{D})$. As $f$ is continuous on a compact set, it is uniformly continuous and we thus have $f(\tilde{\boldsymbol{x}}) - f(\boldsymbol{x}^*) \leq \omega(\delta \operatorname{diam}(\mathcal{D}))$ with $\omega$ its modulus of continuity function.

**Adaptive $\delta$:** The second variant to solve $\min_{\boldsymbol{x} \in \mathcal{D}} f(\boldsymbol{x})$ iteratively perform FW steps over $\mathcal{D}_\delta$, but also decreases $\delta$ adaptively. The update schedule for $\delta$ is given in Alg. 1 and is motivated by the convergence proof. The idea is to ensure that the FW gap over $\mathcal{D}_\delta$ is always at least half the FW gap over $\mathcal{D}$, relating the progress over $\mathcal{D}_\delta$ with the one over $\mathcal{D}$. It turns out that FW-gap-$\mathcal{D}_\delta = (1 - \delta)$FW-gap-$\mathcal{D} + \delta \cdot g_u(\boldsymbol{x}^{(k)})$, where the "uniform gap" $g_u(\boldsymbol{x}^{(k)})$ quantifies the decrease of the function when contracting towards $\boldsymbol{u}_0$. When $g_u(\boldsymbol{x}^{(k)})$ is negative and large compared to the FW gap, we need to shrink $\delta$ (see step 5 in Alg. 1) to ensure that the $\delta$-modified direction is a sufficient descent direction. We can show that the algorithm converges to the global solution as follows:

**Theorem 3.5** (Global convergence for adaptive-$\delta$ variant over $\mathcal{D}$). *For a function* $f$ *satisfying the properties in Prob. 3.2 and Prop. 3.3, the sub-optimality of the iterates obtained by running the FW updates over* $\mathcal{D}_\delta$ *with* $\delta$ *updated according to Alg. 1 is bounded as:*

$$f(\boldsymbol{x}^{(k)}) - f(\boldsymbol{x}^*) \leq O\left(k^{-\frac{1}{p+1}}\right).$$

A full proof with a precise rate and constants is given in App. D. The sub-optimality $h_k := f(\boldsymbol{x}^{(k)}) - f(\boldsymbol{x}^*)$ traverses three stages with an overall rate as above. The updates to $\delta^{(k)}$ as in Alg. 1 enable us

---

**Algorithm 2:** Approximate marginal inference over $\mathcal{M}$ (solving (1)). Here $f$ is the negative TRW objective.

---

1: Function **TRW-Barrier-FW**($\boldsymbol{\rho}^{(0)}, \epsilon, \delta^{(\text{init})}, \boldsymbol{u}_0$):
2: **Inputs:** Edge-appearance probabilities $\boldsymbol{\rho}^{(0)}$, $\delta^{(\text{init})} \leq \frac{1}{4}$ initial contraction of polytope, inner loop
    stopping criterion $\epsilon$, fixed reference point $\boldsymbol{u}_0$ in the interior of $\mathcal{M}$. Let $\delta^{(-1)} = \delta^{(\text{init})}$.
3: Let $V := \{\boldsymbol{u}_0\}$ (visited vertices), $\boldsymbol{x}^{(0)} = \boldsymbol{u}_0$    (Initialize the algorithm at the uniform distribution)
4: **for** $i = 0 \ldots$ **MAX_RHO_ITS do** {*FW outer loop to optimize $\boldsymbol{\rho}$ over $\mathbb{T}$*}
5:    **for** $k = 0 \ldots$ **MAXITS do** {*FCFW inner loop to optimize $\boldsymbol{x}$ over $\mathcal{M}$*}
6:       Let $\tilde{\theta} = \nabla f(\boldsymbol{x}^{(k)}; \vec{\boldsymbol{\theta}}, \boldsymbol{\rho}^{(i)})$   *(Compute gradient)*
7:       Let $\boldsymbol{s}^{(k)} \in \arg\min_{\boldsymbol{v} \in \mathcal{M}} \langle \tilde{\theta}, \boldsymbol{v} \rangle$   *(Run MAP solver to compute FW vertex)*
8:       Compute $g(\boldsymbol{x}^{(k)}) = \langle -\tilde{\theta}, \boldsymbol{s}^{(k)} - \boldsymbol{x}^{(k)} \rangle$   *(Inner loop FW duality gap)*
9:       **if** $g(\boldsymbol{x}^{(k)}) \leq \epsilon$ **then**
10:          **break** FCFW inner loop   *($\boldsymbol{x}^{(k)}$ is $\epsilon$-optimal)*
11:       **end if**
12:       $\delta^{(k)} = \delta^{(k-1)}$   *(For Adaptive-$\delta$: Run Alg. 1 to modify $\delta$)*
13:       Let $\boldsymbol{s}_{(\delta)}^{(k)} = (1 - \delta^{(k)})\boldsymbol{s}^{(k)} + \delta^{(k)}\boldsymbol{u}_0$ and $\boldsymbol{d}_{(\delta)}^{(k)} = \boldsymbol{s}_{(\delta)}^{(k)} - \boldsymbol{x}^{(k)}$   *($\delta$-contracted quantities)*
14:       $\boldsymbol{x}^{(k+1)} = \arg\min\{f(\boldsymbol{x}^{(k)} + \gamma\,\boldsymbol{d}_{(\delta)}^{(k)}) : \gamma \in [0,1]\}$   *(FW step with line search)*
15:       Update correction polytope: $V := V \cup \{\boldsymbol{s}^{(k)}\}$
16:       $\boldsymbol{x}^{(k+1)} := \textbf{CORRECTION}(\boldsymbol{x}^{(k+1)}, V, \delta^{(k)}, \boldsymbol{\rho}^{(i)})$   *(optional: correction step)*
17:       $\boldsymbol{x}^{(k+1)}, V_{\text{search}} := \textbf{LOCALSEARCH}(\boldsymbol{x}^{(k+1)}, \boldsymbol{s}^{(k)}, \delta^{(k)}, \boldsymbol{\rho}^{(i)})$   *(optional: fast MAP solver)*
18:       Update correction polytope (with vertices from **LOCALSEARCH**): $V := V \cup \{V_{\text{search}}\}$
19:    **end for**
20:    $\boldsymbol{\rho}^v \leftarrow \textbf{minSpanTree}(\textbf{edgesMI}(\boldsymbol{x}^{(k)}))$   *(FW vertex of the spanning tree polytope)*
21:    $\boldsymbol{\rho}^{(i+1)} \leftarrow \boldsymbol{\rho}^{(i)} + (\frac{i}{i+2})(\boldsymbol{\rho}^v - \boldsymbol{\rho}^{(i)})$   *(Fixed step-size schedule FW update for $\boldsymbol{\rho}$ kept in $\text{relint}(\mathbb{T})$)*
22:    $\boldsymbol{x}^{(0)} \leftarrow \boldsymbol{x}^{(k)}, \quad \delta^{(-1)} \leftarrow \delta^{(k-1)}$   *(Re-initialize for FCFW inner loop)*
23:    If $i <$ **MAX_RHO_ITS** then $\boldsymbol{x}^{(0)} = \textbf{CORRECTION}(\boldsymbol{x}^{(0)}, V, \delta^{(-1)}, \boldsymbol{\rho}^{(i+1)})$
24: **end for**
25: **return** $\boldsymbol{x}^{(0)}$ and $\boldsymbol{\rho}^{(i)}$

---

to (1) upper bound the duality gap over $\mathcal{D}$ as a function of the duality gap in $\mathcal{D}_\delta$ and (2) lower bound the value of $\delta^{(k)}$ as a function of $h_k$. Applying the standard Descent Lemma with the Lipschitz constant on the gradient of the form $L\delta^{-p}$ (Prop. 3.3), and replacing $\delta^{(k)}$ by its bound in $h_k$, we get the recurrence: $h_{k+1} \leq h_k - Ch_k^{p+2}$. Solving this gives us the desired bound.

**Application to the TRW Objective:** $\min_{\vec{\boldsymbol{\mu}} \in \mathcal{M}} -\text{TRW}(\vec{\boldsymbol{\mu}}; \vec{\boldsymbol{\theta}}, \boldsymbol{\rho})$ is akin to $\min_{\boldsymbol{x} \in \mathcal{D}} f(\boldsymbol{x})$ and the (strong) convexity of $-\text{TRW}(\vec{\boldsymbol{\mu}}; \vec{\boldsymbol{\theta}}, \boldsymbol{\rho})$ has been previously shown (Wainwright et al., 2005; London et al., 2015). The gradient of the TRW objective is Lipschitz continuous over $\mathcal{M}_\delta$ since all marginals are strictly positive. Its growth for Prop. 3.3 can be bounded with $p = 1$ as we show in App. E.1. This gives a rate of convergence of $O(k^{-1/2})$ for the adaptive-$\delta$ variant, which interestingly is a typical rate for non-smooth convex optimization. The hidden constant is of the order $O(\|\theta\| \cdot |V|)$. The modulus of continuity $\omega$ for the TRW objective is close to linear (it is almost a Lipschitz function), and its constant is instead of the order $O(\|\theta\| + |V|)$.

## 4 Algorithm

Alg. 2 describes the pseudocode for our proposed algorithm to do marginal inference with $\text{TRW}(\vec{\boldsymbol{\mu}}; \vec{\boldsymbol{\theta}}, \boldsymbol{\rho})$. **minSpanTree** finds the minimum spanning tree of a weighted graph, and **edgesMI**($\vec{\boldsymbol{\mu}}$) computes the mutual information of edges of $G$ from the pseudomarginals in $\vec{\boldsymbol{\mu}}$[2] (to perform FW updates over $\boldsymbol{\rho}$ as in Alg. 2 in Wainwright et al. (2005)). It is worthwhile to note that our approach uses three levels of Frank-Wolfe: (1) for the (tightening) optimization of $\boldsymbol{\rho}$ over $\mathbb{T}$, (2) to perform approximate marginal inference, i.e for the optimization of $\vec{\boldsymbol{\mu}}$ over $\mathcal{M}$, and (3) to perform the correction steps (lines 16 and 23). We detail a few heuristics that aid practicality.

**Fast Local Search:** Fast methods for MAP inference such as Iterated Conditional Modes (Besag, 1986) offer a cheap, low cost alternative to a more expensive combinatorial MAP solver. We

warm start the ICM solver with the last found vertex $s^{(k)}$ of the marginal polytope. The subroutine **LOCALSEARCH** (Alg. 6 in Appendix) performs a fixed number of FW updates to the pseudo-marginals using ICM as the (approximate) MAP solver.

**Re-optimizing over the Vertices of $\mathcal{M}$ (FCFW algorithm):** As the iterations of FW progress, we keep track of the vertices of the marginal polytope found by Alg. 2 in the set $V$. We make use of these vertices in the **CORRECTION** subroutine (Alg. 5 in Appendix) which re-optimizes the objective function over (a contraction of) the convex hull of the elements of $V$ (called the correction polytope). $x^{(0)}$ in Alg. 2 is initialized to the uniform distribution which is guaranteed to be in $\mathcal{M}$ (and $\mathcal{M}_\delta$). After updating $\rho$, we set $x^{(0)}$ to the approximate minimizer in the correction polytope. The intuition is that changing $\rho$ by a small amount may not substantially modify the optimal $x^*$ (for the new $\rho$) and that the new optimum might be in the convex hull of the vertices found thus far. If so, **CORRECTION** will be able to find it without resorting to any additional MAP calls. This encourages the MAP solver to search for new, unique vertices instead of rediscovering old ones.

**Approximate MAP Solvers:** We can swap out the exact MAP solver with an approximate MAP solver. The primal objective plus the (approximate) duality gap may no longer be an upper bound on the log-partition function (black-box MAP solvers could be considered to optimize over an inner bound to the marginal polytope). Furthermore, the gap over $\mathcal{D}$ may be negative if the approximate MAP solver fails to find a direction of descent. Since adaptive-$\delta$ requires that the gap be positive in Alg. 1, we take the max over the last gap obtained over the correction polytope (which is always non-negative) and the computed gap over $\mathcal{D}$ as a heuristic.

Theoretically, one could get similar convergence rates as in Thm. 3.4 and 3.5 using an approximate MAP solver that has a multiplicative guarantee on the gap (line 8 of Alg. 2), as was done previously for FW-like algorithms (see, e.g., Thm. C.1 in Lacoste-Julien et al. (2013)). With an $\epsilon$-additive error guarantee on the MAP solution, one can prove similar rates up to a suboptimality error of $\epsilon$. Even if the approximate MAP solver does not provide an approximation guarantee, if it returns an *upper bound* on the value of the MAP assignment (as do branch-and-cut solvers for integer linear programs, or Sontag et al. (2008)), one can use this to obtain an upper bound on $\log Z$ (see App. J).

## 5   Experimental Results

**Setup:** The L1 error in marginals is computed as: $\zeta_\mu := \frac{1}{N} \sum_{i=1}^{N} |\mu_i(1) - \mu_i^*(1)|$. When using exact MAP inference, the error in $\log Z$ (denoted $\zeta_{\log Z}$) is computed by adding the duality gap to the primal (since this guarantees us an upper bound). For approximate MAP inference, we plot the primal objective. We use a non-uniform initialization of $\rho$ computed with the Matrix Tree Theorem (Sontag and Jaakkola, 2007; Koo et al., 2007). We perform 10 updates to $\rho$, optimize $\vec{\mu}$ to a duality gap of $0.5$ on $\mathcal{M}$, and always perform correction steps. We use **LOCALSEARCH** only for the real-world instances. We use the implementation of TRBP and the Junction Tree Algorithm (to compute exact marginals) in libDAI (Mooij, 2010). Unless specified, we compute marginals by optimizing the TRW objective using the adaptive-$\delta$ variant of the algorithm (denoted in the figures as $M_\delta$).

**MAP Solvers:** For approximate MAP, we run three solvers in parallel: QPBO (Kolmogorov and Rother, 2007; Boykov and Kolmogorov, 2004), TRW-S (Kolmogorov, 2006) and ICM (Besag, 1986) using OpenGM (Andres et al., 2012) and use the result that realizes the highest energy. For exact inference, we use Gurobi Optimization (2015) or toulbar2 (Allouche et al., 2010).

**Test Cases:** All of our test cases are on binary pairwise MRFs. (1) *Synthetic 10 nodes cliques*: Same setup as Sontag and Jaakkola (2007, Fig. 2), with 9 sets of 100 instances each with coupling strength drawn from $\mathcal{U}[-\theta, \theta]$ for $\theta \in \{0.5, 1, 2, \ldots, 8\}$. (2) *Synthetic Grids*: 15 trials with $5 \times 5$ grids. We sample $\theta_i \sim \mathcal{U}[-1, 1]$ and $\theta_{ij} \in [-4, 4]$ for nodes and edges. The potentials were $(-\theta_i, \theta_i)$ for nodes and $(\theta_{ij}, -\theta_{ij}; -\theta_{ij}, \theta_{ij})$ for edges. (3) *Restricted Boltzmann Machines (RBMs)*: From the Probabilistic Inference Challenge 2011.[3] (4) *Horses*: Large ($N \approx 12000$) MRFs representing images from the Weizmann Horse Data (Borenstein and Ullman, 2002) with potentials learned by Domke (2013). (5) *Chinese Characters*: An image completion task from the KAIST Hanja2 database, compiled in OpenGM by Andres et al. (2012). The potentials were learned using Decision Tree Fields (Nowozin et al., 2011). The MRF is not a grid due to skip edges that tie nodes at various offsets. The potentials are a combination of submodular and supermodular and therefore a harder task for inference algorithms.

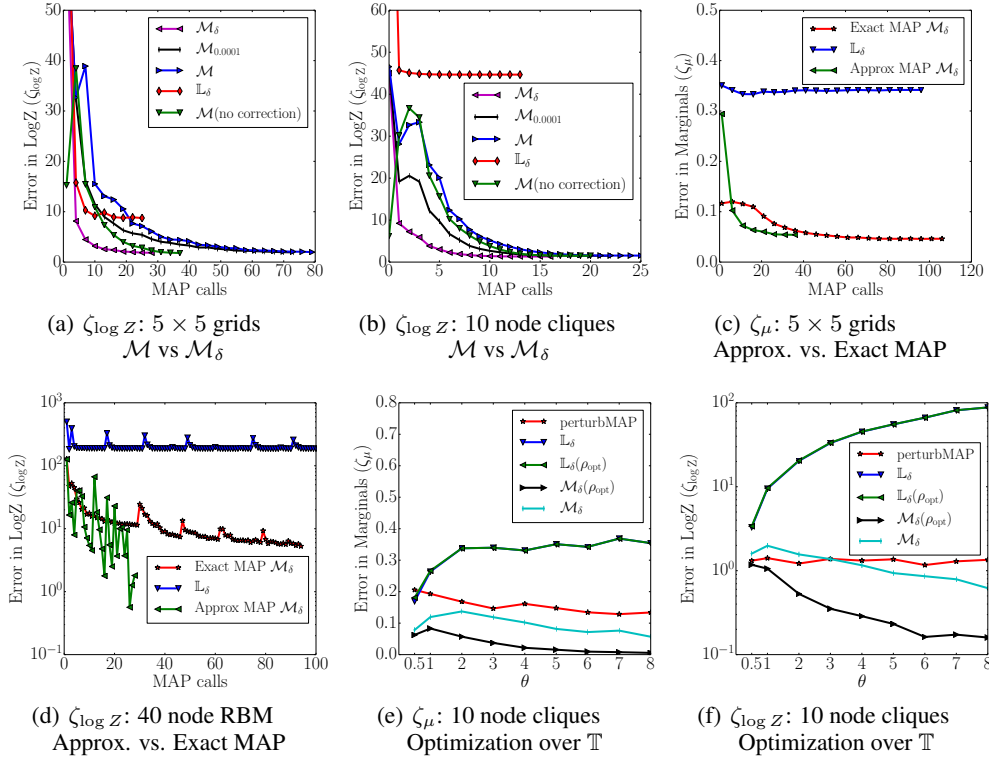

(a) $\zeta_{\log Z}$: $5 \times 5$ grids
$\mathcal{M}$ vs $\mathcal{M}_\delta$

(b) $\zeta_{\log Z}$: 10 node cliques
$\mathcal{M}$ vs $\mathcal{M}_\delta$

(c) $\zeta_\mu$: $5 \times 5$ grids
Approx. vs. Exact MAP

(d) $\zeta_{\log Z}$: 40 node RBM
Approx. vs. Exact MAP

(e) $\zeta_\mu$: 10 node cliques
Optimization over $\mathbb{T}$

(f) $\zeta_{\log Z}$: 10 node cliques
Optimization over $\mathbb{T}$

Figure 1: Synthetic Experiments: In Fig. 1(c) & 1(d), we unravel MAP calls across updates to $\rho$. Fig. 1(d) corresponds to a single RBM (not an aggregate over trials) where for "Approx MAP" we plot the absolute error between the primal objective and $\log Z$ (not guaranteed to be an upper bound).

### On the Optimization of $\mathcal{M}$ versus $\mathcal{M}_\delta$

We compare the performance of Alg. 2 on optimizing over $\mathcal{M}$ (with and without correction), optimizing over $\mathcal{M}_\delta$ with fixed-$\delta = 0.0001$ (denoted $M_{0.0001}$) and optimizing over $\mathcal{M}_\delta$ using the adaptive-$\delta$ variant. These plots are averaged across all the trials for the *first* iteration of optimizing over $\mathbb{T}$. We show error as a function of the number of MAP calls since this is the bottleneck for large MRFs. Fig. 1(a), 1(b) depict the results of this optimization aggregated across trials. We find that all variants settle on the same average error. The adaptive $\delta$ variant converges faster on average followed by the fixed $\delta$ variant. Despite relatively quick convergence for $\mathcal{M}$ with no correction on the grids, we found that correction was crucial to reducing the number of MAP calls in subsequent steps of inference after updates to $\rho$. As highlighted earlier, correction steps on $\mathcal{M}$ (in blue) worsen convergence, an effect brought about by iterates wandering too close to the boundary of $\mathcal{M}$.

### On the Applicability of Approximate MAP Solvers

**Synthetic Grids:** Fig. 1(c) depicts the accuracy of approximate MAP solvers versus exact MAP solvers aggregated across trials for $5 \times 5$ grids. The results using approximate MAP inference are competitive with those of exact inference, even as the optimization is tightened over $\mathbb{T}$. This is an encouraging and non-intuitive result since it indicates that one can achieve high quality marginals through the use of relatively cheaper approximate MAP oracles.

**RBMs:** As in Salakhutdinov (2008), we observe for RBMs that the bound provided by $TRW(\vec{\boldsymbol{\mu}}; \vec{\boldsymbol{\theta}}, \boldsymbol{\rho})$ over $\mathbb{L}_\delta$ is loose and does not get better when optimizing over $\mathbb{T}$. As Fig. 1(d) depicts for a single RBM, optimizing over $\mathcal{M}_\delta$ realizes significant gains in the upper bound on $\log Z$ which improves with updates to $\rho$. The gains are preserved with the use of the approximate MAP solvers. Note that there are also fast approximate MAP solvers specifically for RBMs (Wang et al., 2014).

**Horses:** See Fig. 2 (right). The models are close to submodular and the local relaxation is a good approximation to the marginal polytope. Our marginals are visually similar to those obtained by TRBP and our algorithm is able to scale to large instances by using approximate MAP solvers.

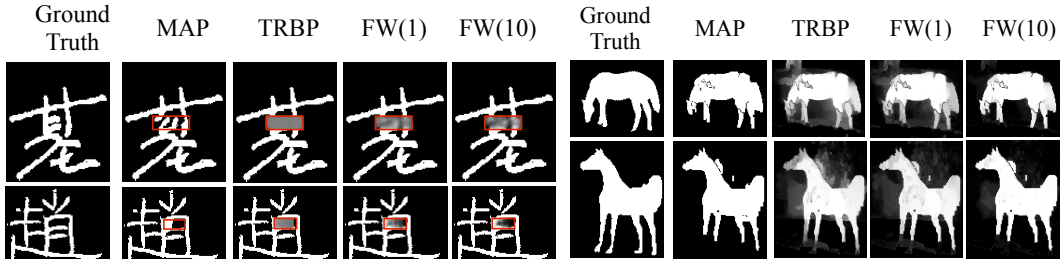

Figure 2: Results on real world test cases. FW(i) corresponds to the final marginals at the $i$th iteration of optimizing $\rho$. The area highlighted on the Chinese Characters depicts the region of uncertainty.

### On the Importance of Optimizing over $\mathbb{T}$

**Synthetic Cliques:** In Fig. 1(e), 1(f), we study the effect of tightening over $\mathbb{T}$ against coupling strength $\theta$. We consider the $\zeta_\mu$ and $\zeta_{\log Z}$ obtained for the final marginals before updating $\rho$ (step 19) and compare to the values obtained after optimizing over $\mathbb{T}$ (marked with $\rho_{opt}$). The optimization over $\mathbb{T}$ has little effect on TRW optimized over $\mathbb{L}_\delta$. For optimization over $\mathcal{M}_\delta$, updating $\rho$ realizes better marginals and bound on $\log Z$ (over and above those obtained in Sontag and Jaakkola (2007)).

**Chinese Characters:** Fig. 2 (left) displays marginals across iterations of optimizing over $\mathbb{T}$. The submodular and supermodular potentials lead to frustrated models for which $\mathbb{L}_\delta$ is very loose, which results in TRBP obtaining poor results.[4] Our method produces reasonable marginals even before the first update to $\rho$, and these improve with tightening over $\mathbb{T}$.

### Related Work for Marginal Inference with MAP Calls

Hazan and Jaakkola (2012) estimate $\log Z$ by averaging MAP estimates obtained on randomly perturbed inflated graphs. Our implementation of the method performed well in approximating $\log Z$ but the marginals (estimated by fixing the value of each random variable and estimating $\log Z$ for the resulting graph) were less accurate than our method (Fig. 1(e), 1(f)).

## 6 Discussion

We introduce the first provably convergent algorithm for the TRW objective over the marginal polytope, under the assumption of exact MAP oracles. We quantify the gains obtained both from marginal inference over $\mathcal{M}$ *and* from tightening over the spanning tree polytope. We give heuristics that improve the scalability of Frank-Wolfe when used for marginal inference. The runtime cost of iterative MAP calls (a reasonable rule of thumb is to assume an approximate MAP call takes roughly the same time as a run of TRBP) is worthwhile particularly in cases such as the Chinese Characters where $\mathbb{L}$ is loose. Specifically, our algorithm is appropriate for domains where marginal inference is hard but there exist efficient MAP solvers capable of handling non-submodular potentials. Code is available at `https://github.com/clinicalml/fw-inference`.

Our work creates a flexible, modular framework for optimizing a broad class of variational objectives, not simply TRW, with guarantees of convergence. We hope that this will encourage more research on building better entropy approximations. The framework we adopt is more generally applicable to optimizing functions whose gradients tend to infinity at the boundary of the domain.

Our method to deal with gradients that diverge at the boundary bears resemblance to barrier functions used in interior point methods insofar as they bound the solution away from the constraints. Iteratively decreasing $\delta$ in our framework can be compared to decreasing the strength of the barrier, enabling the iterates to get closer to the facets of the polytope, although its worthwhile to note that we have an *adaptive* method of doing so.

## Acknowledgements

RK and DS gratefully acknowledge the support of the DARPA Probabilistic Programming for Advancing Machine Learning (PPAML) Program under AFRL prime contract no. FA8750-14-C-0005.

## Footnotes

[1] I.e. $\|\nabla f(\boldsymbol{x}) - \nabla f(\boldsymbol{x}')\|_* \leq L \|\boldsymbol{x} - \boldsymbol{x}'\|$ for $\boldsymbol{x}, \boldsymbol{x}' \in \mathcal{D}$. Notice that the dual norm $\|\cdot\|_*$ is needed here.

[2]The component $ij$ has value $H(\boldsymbol{\mu}_i) + H(\boldsymbol{\mu}_j) - H(\boldsymbol{\mu}_{ij})$.

[3] http://www.cs.huji.ac.il/project/PASCAL/index.php

[4]We run TRBP for 1000 iterations using damping = 0.9; the algorithm converges with a max norm difference between consecutive iterates of 0.002. Tightening over $\mathbb{T}$ did not significantly change the results of TRBP.

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
