[Supplementary Material]

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

[5]Note that without a strong convexity assumption on $f$, the optimum over $\mathcal{D}_\delta$, $\boldsymbol{x}^*_{(\delta)}$, could be quite far from the optimum over $\mathcal{D}$, $\boldsymbol{x}^*$, which is why we need to construct this alternative close point to $\boldsymbol{x}^*$.

[6]Note that on the other hand, $g_u(\boldsymbol{x})$ might go to infinity as $\boldsymbol{x}$ gets close to the boundary of $\mathcal{D}$ as the gradient of $f$ is allowed to be unbounded. Fortunately, we only need an upper bound on $-g_u$, not a lower bound.

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

# A   Preliminaries

## A.1   Summary of Supplementary Material

The supplementary material is divided into two parts:

(1) The first part is dedicated to the exposition of the theoretical results presented in the main paper. Section B details the variants of the Frank-Wolfe algorithm that we used and analyzed. Section C gives the proof to Theorem 3.4 (fixed $\delta$) while Section D gives the proof to Theorem 3.5 (adaptive $\delta$). Finally, Section E applies the convergence theorem to the TRW objective and investigates the relevant constants.

(2) The remainder of the supplementary material provides more information about the experimental setup as well as additional experimental results.

## A.2   Descent Lemma

The following descent lemma is proved in Bertsekas (1999) (Prop. A24) and is standard for any convergence proof of first order methods. We provide a proof here for completeness. It also highlights the origin of the requirement that we use dual norm pairings between $\boldsymbol{x}$ and the gradient of $f(\boldsymbol{x})$ (because of the generalized Cauchy-Schwartz inequality).

**Lemma A.1.** *Descent Lemma*

*Let $\boldsymbol{x}_\gamma := \boldsymbol{x} + \gamma \boldsymbol{d}$ and suppose that $f$ is continuously differentiable on the line segment from $\boldsymbol{x}$ to $\boldsymbol{x}_{\gamma\max}$ for some $\gamma_{\max} > 0$. Suppose that $L = \sup_{\alpha \in ]0,\gamma_{\max}]} \frac{||\nabla f(\boldsymbol{x}+\alpha\boldsymbol{d}) - \nabla f(\boldsymbol{x})||_*}{||\alpha\boldsymbol{d}||}$ is finite, then we have:*

$$f(\boldsymbol{x}_\gamma) \le f(\boldsymbol{x}) + \gamma \langle \nabla f(\boldsymbol{x}), \boldsymbol{d} \rangle + \frac{\gamma^2}{2} L ||\boldsymbol{d}||^2, \quad \forall \gamma \in [0, \gamma_{\max}]. \tag{5}$$

*Proof.* Let $0 < \gamma \le \gamma_{\max}$. Denoting $l(\alpha) = f(\boldsymbol{x} + \alpha\boldsymbol{d})$, we have that:

$$
\begin{aligned}
f(\boldsymbol{x}_\gamma) - f(\boldsymbol{x}) &= l(\gamma) - l(0) \\
&= \int_0^\gamma \nabla_\alpha l(\alpha) d\alpha \\
&= \int_0^\gamma \langle \boldsymbol{d}, \nabla f(\boldsymbol{x} + \alpha\boldsymbol{d}) \rangle d\alpha \\
&= \int_0^\gamma \langle \boldsymbol{d}, \nabla f(\boldsymbol{x}) \rangle d\alpha + \int_0^\gamma \boldsymbol{d}^T (\nabla f(\boldsymbol{x} + \alpha\boldsymbol{d}) - \nabla f(\boldsymbol{x})) d\alpha \\
&\le \int_0^\gamma \langle \boldsymbol{d}, \nabla f(\boldsymbol{x}) \rangle d\alpha + \left| \int_0^\gamma \boldsymbol{d}^T (\nabla f(\boldsymbol{x} + \alpha\boldsymbol{d}) - \nabla f(\boldsymbol{x})) \right| d\alpha \\
&\le \int_0^\gamma \langle \boldsymbol{d}, \nabla f(\boldsymbol{x}) \rangle d\alpha + \int_0^\gamma ||\boldsymbol{d}||\ ||\nabla f(\boldsymbol{x} + \alpha\boldsymbol{d}) - \nabla f(\boldsymbol{x})||_* d\alpha \\
&= \gamma \langle \boldsymbol{d}, \nabla f(\boldsymbol{x}) \rangle + \int_0^\gamma ||\boldsymbol{d}||\ \frac{||\nabla f(\boldsymbol{x} + \alpha\boldsymbol{d}) - \nabla f(\boldsymbol{x})||_*}{\alpha ||\boldsymbol{d}||} \alpha ||\boldsymbol{d}|| d\alpha \\
&\le \gamma \langle \boldsymbol{d}, \nabla f(\boldsymbol{x}) \rangle + \int_0^\gamma ||\boldsymbol{d}||\ L ||\boldsymbol{d}|| \alpha\, d\alpha \\
&= \gamma \langle \boldsymbol{d}, \nabla f(\boldsymbol{x}) \rangle + \frac{L}{2} \gamma^2 ||\boldsymbol{d}||^2
\end{aligned}
$$

Rearranging terms, we get the desired bound. □

# B   Frank-Wolfe Algorithms

In this section, we present the various algorithms that we use to do fully corrective Frank-Wolfe (FCFW) with adaptive contractions over the domain $\mathcal{D}$, as was done in our experiments.

## B.1 Overview of the Modified Frank-Wolfe Algorithm (FW with Away Steps)

To implement the approximate correction steps in the fully corrective Frank-Wolfe (FCFW) algorithm, we use the Frank-Wolfe algorithm with away steps (Wolfe, 1970), also known as the modified Frank-Wolfe (MFW) algorithm (Guélat and Marcotte, 1986). We give pseudo-code for MFW in Algorithm 3 (taken from (Lacoste-Julien and Jaggi, 2015)). This variant of Frank-Wolfe adds the possibility to do an "away step" (see step 5 in Algorithm 3) in order to avoid the zig zagging phenomenon that slows down Frank-Wolfe when the solution is close to the boundary of the polytope. For a strongly convex objective (with Lipschitz continuous gradient), the MFW was known to have asymptotic linear convergence (Guélat and Marcotte, 1986) and its global linear convergence rate was shown recently (Lacoste-Julien and Jaggi, 2015), accelerating the slow general sublinear rate of Frank-Wolfe. When performing a correction over the convex hull over a (somewhat small) set of vertices of $\mathcal{D}_\delta$, this convergence difference was quite significant in our experiments (MFW converging in a small number of iterations to do an approximate correction vs. FW taking hundreds of iterations to reach a similar level of accuracy). We note that the TRW objective is strongly convex when all the edge probabilities are non-zero (Wainwright et al., 2005); and that it has Lipschitz gradient over $\mathcal{D}_\delta$ (but not $\mathcal{D}$).

The gap computed in step 6 of Algorithm 3 is non-standard; it is a sufficient condition to ensure the global linear convergence of the outer FCFW algorithm when using Algorithm 3 as a subroutine to implement the approximate correction step. See Lacoste-Julien and Jaggi (2015) for more details.

The MFW algorithm requires more bookkeeping than standard FW: in addition to the current iterate $\boldsymbol{x}^{(k)}$, it also maintains both the active set $\mathcal{S}^{(k)}$ (to search for the "away vertex") as well as the barycentric coordinates $\boldsymbol{\alpha}^{(k)}$ (to know what are the away step-sizes that ensure feasibility – see step 13) i.e. $\boldsymbol{x}^{(k)} = \sum_{\boldsymbol{v} \in \mathcal{S}^{(k)}} \alpha_{\boldsymbol{v}}^{(k)} \boldsymbol{v}$.

---

**Algorithm 3:** Modified Frank-Wolfe algorithm (FW with Away Steps) – used for approximate correction

1: Function **MFW**$(\boldsymbol{x}^{(0)}, \boldsymbol{\alpha}^{(0)}, \mathcal{V}, \epsilon)$ to optimize over $\text{conv}(\mathcal{V})$:
2: **Inputs:** Set of atoms $\mathcal{V}$, starting point $\boldsymbol{x}^{(0)} = \sum_{\boldsymbol{v} \in \mathcal{S}^{(0)}} \alpha_{\boldsymbol{v}}^{(0)} \boldsymbol{v}$ where $\mathcal{S}^{(0)}$ is active set and $\boldsymbol{\alpha}^{(0)}$ the active coordinates, stopping criterion $\epsilon$.
3: **for** $k = 0 \dots K$ **do**
4:   Let $\boldsymbol{s}_k \in \underset{\boldsymbol{v} \in \mathcal{V}}{\arg\min} \langle \nabla f(\boldsymbol{x}^{(k)}), \boldsymbol{v} \rangle$ and $\boldsymbol{d}_k^{\text{FW}} := \boldsymbol{s}_k - \boldsymbol{x}^{(k)}$    *(the FW direction)*
5:   Let $\boldsymbol{v}_k \in \underset{\boldsymbol{v} \in \mathcal{S}^{(k)}}{\arg\max} \langle \nabla f(\boldsymbol{x}^{(k)}), \boldsymbol{v} \rangle$ and $\boldsymbol{d}_k^{\text{A}} := \boldsymbol{x}^{(k)} - \boldsymbol{v}_k$    *(the away direction)*
6:   $g_k^{\text{pFW}} := \left\langle -\nabla f(\boldsymbol{x}^{(k)}), \boldsymbol{d}_k^{\text{FW}} + \boldsymbol{d}_k^{\text{A}} \right\rangle$    *(stringent gap is FW + away gap to work better for FCFW)*
7:   **if** $g_k^{\text{pFW}} \leq \epsilon$ **then**
8:     **return** $\boldsymbol{x}^{(k)}, \boldsymbol{\alpha}^{(k)}, \mathcal{S}^{(k)}$.
9:   **else**
10:     **if** $\left\langle -\nabla f(\boldsymbol{x}^{(k)}), \boldsymbol{d}_k^{\text{FW}} \right\rangle \geq \left\langle -\nabla f(\boldsymbol{x}^{(k)}), \boldsymbol{d}_k^{\text{A}} \right\rangle$ **then**
11:       $\boldsymbol{d}_k := \boldsymbol{d}_k^{\text{FW}}$, and $\gamma_{\max} := 1$    *(choose the FW direction)*
12:     **else**
13:       $\boldsymbol{d}_k := \boldsymbol{d}_k^{\text{A}}$, and $\gamma_{\max} := \frac{\alpha_{\boldsymbol{v}_k}}{(1 - \alpha_{\boldsymbol{v}_k})}$    *(choose away direction; maximum feasible step-size)*
14:     **end if**
15:     Line-search: $\gamma_k \in \underset{\gamma \in [0, \gamma_{\max}]}{\arg\min} f\left(\boldsymbol{x}^{(k)} + \gamma \boldsymbol{d}_k\right)$
16:     Update $\boldsymbol{x}^{(k+1)} := \boldsymbol{x}^{(k)} + \gamma_k \boldsymbol{d}_k$
17:     Update coordinates $\boldsymbol{\alpha}^{(k+1)}$ accordingly (see Lacoste-Julien and Jaggi (2015)).
18:     Update $\mathcal{S}^{(k+1)} := \{\boldsymbol{v} \ s.t. \ \alpha_{\boldsymbol{v}}^{(k+1)} > 0\}$
19:   **end if**
20: **end for**

---

## B.2 Fully Corrective Frank-Wolfe (FCFW) with Adaptive-$\delta$

We give in Algorithm 4 the pseudo-code to perform fully corrective Frank-Wolfe optimization over $\mathcal{D}$ by iteratively optimizing over $\mathcal{D}_\delta$ with adaptive-$\delta$ updates. If $\delta$ is kept constant (skipping step 10), then Algorithm 4 implements the fixed $\delta$ variant over $\mathcal{D}_\delta$. We describe the algorithm as maintaining the correction set of atoms $V^{(k+1)}$ over $\mathcal{D}$ (rather than $\mathcal{D}_\delta$), as $\delta$ is constantly changing. One can easily move back and forth between $V^{(k+1)}$ and its contraction $V_\delta = (1 - \delta^{(k)})V^{(k+1)} + \delta^{(k)} \boldsymbol{u}_0$, and so we note that an efficient implementation might work with either representation cheaply (for example, by storing only $V^{(k+1)}$ and $\delta$, not the perturbed version of the correction polytope). The approximate correction over $V_\delta$ is implemented using the MFW algorithm described in Algorithm 3, which requires a barycentric representation $\boldsymbol{\alpha}^{(k)}$ of the current iter-

ate $\boldsymbol{x}^{(k)}$ over the correction polytope $V_\delta$. Our notation in Algorithm 4 uses the elements of $\mathcal{V}$ as indices, rather than their contracted version; that is, we maintain the property that $\boldsymbol{x}^{(k)} = \sum_{\boldsymbol{v} \in \mathcal{V}} \alpha_{\boldsymbol{v}}^{(k)}[(1 - \delta^{(k)})\boldsymbol{v} + \delta^{(k)}\boldsymbol{u}_0]$. As $V_\delta$ changes when $\delta$ changes, we need to update the barycentric representation of $\boldsymbol{x}^{(k)}$ accordingly – this is done in step 11 with the following equation. Suppose that we decrease $\delta$ to $\delta'$. Then the old coordinates $\boldsymbol{\alpha}$ can be updated to new coordinates $\boldsymbol{\alpha}'$ for the new contraction polytope as follows:

$$
\begin{aligned}
\alpha'_{\boldsymbol{v}} &= \alpha_{\boldsymbol{v}} \frac{1 - \delta}{1 - \delta'} \quad \text{for} \quad \boldsymbol{v} \in \mathcal{V} \setminus \{\boldsymbol{u}_0\}, \\
\alpha'_{\boldsymbol{u}_0} &= 1 - \sum_{\boldsymbol{v} \neq \boldsymbol{u}_0} \alpha'_{\boldsymbol{v}}.
\end{aligned}
\tag{6}
$$

This ensures that $\sum_{\boldsymbol{v}} \alpha_{\boldsymbol{v}} \boldsymbol{v}_{(\delta)} = \sum_{\boldsymbol{v}} \alpha'_{\boldsymbol{v}} \boldsymbol{v}_{(\delta')}$, where $\boldsymbol{v}_{(\delta)} := (1 - \delta)\boldsymbol{v} + \delta \boldsymbol{u}_0$, and that the coordinates form a valid convex combination (assuming that $\delta' \leq \delta$), as can be readily verified.

---

**Algorithm 4:** Optimizing $f$ over $\mathcal{D}$ using Fully Corrective Frank-Wolfe (FCFW) with Adaptive-$\delta$ Algorithm.

1:  **FCFW**($\boldsymbol{x}^{(0)}, \mathcal{V}, \epsilon, \delta^{(\text{init})}$)
2:  **Inputs:** Set of atoms $\mathcal{V}$ so that $\mathcal{D} = \text{conv}(\mathcal{V})$, active set $\mathcal{S}^{(0)}$, starting point
     $\boldsymbol{x}^{(0)} = \sum_{\boldsymbol{v} \in \mathcal{S}^{(0)}} \alpha_{\boldsymbol{v}}^{(0)}[(1 - \delta^{(\text{init})})\boldsymbol{v} + \delta^{(\text{init})}\boldsymbol{u}_0]$ where $\boldsymbol{\alpha}^{(0)}$ are the active coordinates, $\delta^{(\text{init})} \leq \frac{1}{4}$
     describes the initial contraction of the polytope, stopping criterion $\epsilon$, $\boldsymbol{u}_0$ is a fixed reference point in the
     relative interior of $\mathcal{D}$.
3:  Let $V^{(0)} := \mathcal{S}^{(0)}$   (optionally, a bigger $V^{(0)}$ could be passed as argument for a warm start),
     $\delta^{(-1)} := \delta^{(\text{init})}$
4:  **for** $k = 0 \ldots K$ **do**
5:     Let $\boldsymbol{s}^{(k)} \in \arg\min_{\boldsymbol{v} \in \mathcal{V}} \left\langle \nabla f(\boldsymbol{x}^{(k)}), \boldsymbol{v} \right\rangle$   *(the FW vertex)*
6:     Compute $g(\boldsymbol{x}^{(k)}) = \langle -\nabla f(\boldsymbol{x}^{(k)}), \boldsymbol{s}^{(k)} - \boldsymbol{x}^{(k)} \rangle$   *(FW gap)*
7:     **if** $g(\boldsymbol{x}^{(k)}) \leq \epsilon$ **then**
8:        **return** $\boldsymbol{x}^{(k)}$
9:     **end if**
10:    Let $\delta^{(k)}$ be $\delta^{(k-1)}$ updated according to Algorithm 1.
11:    Update $\boldsymbol{\alpha}^{(k)}$ accordingly (using (6))
12:    Let $\boldsymbol{s}_{(\delta)}^{(k)} := (1 - \delta^{(k)})\boldsymbol{s}^{(k)} + \delta^{(k)}\boldsymbol{u}_0$
13:    Let $\boldsymbol{d}_k^{\text{FW}} := \boldsymbol{s}_{(\delta)}^{(k)} - \boldsymbol{x}^{(k)}$
14:    Line-search: $\gamma_k \in \arg\min_{\gamma \in [0,1]} f\left(\boldsymbol{x}^{(k)} + \gamma \boldsymbol{d}_k^{\text{FW}}\right)$
15:    Set $\boldsymbol{x}^{(\text{temp})} := \boldsymbol{x}^{(k)} + \gamma_k \boldsymbol{d}_k^{\text{FW}}$   *(initialize correction to the update after a FW step with line search)*
16:    $\boldsymbol{\alpha}^{(\text{temp})} = (1 - \gamma_k)\boldsymbol{\alpha}^{(k)}$
17:    $\alpha_{\boldsymbol{s}^{(k)}}^{(\text{temp})} \leftarrow \alpha_{\boldsymbol{s}^{(k)}}^{(\text{temp})} + \gamma_k$   *(update coordinates according to the FW step)*
18:    Update (non-contracted) correction polytope: $V^{(k+1)} := V^{(k)} \cup \{\boldsymbol{s}^{(k)}\}$
19:    Let $V_\delta = (1 - \delta^{(k)})V^{(k+1)} + \delta^{(k)}\boldsymbol{u}_0$   *(contracted correction polytope)*
20:    $\boldsymbol{x}^{(k+1)}, \boldsymbol{\alpha}^{(k+1)} := \text{MFW}(\boldsymbol{x}^{(\text{temp})}, \boldsymbol{\alpha}^{(\text{temp})}, V_\delta, \epsilon)$   *(approximate correction step on $V_\delta$ using MFW)*
21: **end for**

---

## C   Bounding the Sub-optimality for Fixed $\delta$ Variant

The pseudocode for optimizing over $\mathcal{D}_\delta$ for a fixed $\delta$ is given in Algorithm 4 (by ignoring the step 10 which updates $\delta$). It is stated with a stopping criterion $\epsilon$, but it can alternatively be run for a fixed number of $K$ iterations. The following theorem bounds the suboptimality of the iterates with respect to the true optimum $\boldsymbol{x}^*$ over $\mathcal{D}$. If one can compute the constants in the theorem, one can choose a target contraction amount $\delta$ to guarantee a specific suboptimality of $\epsilon'$; otherwise, one can choose $\delta$ using heuristics. Note that unlike the adaptive-$\delta$ variant, this algorithm does not converge to the true solution as $K \to \infty$ unless $\boldsymbol{x}^*$ happens to belong to $\mathcal{D}_\delta$. But the error can be controlled by choosing $\delta$ small enough.

**Theorem C.1** (Suboptimality bound for fixed-$\delta$ algorithm). *Let $f$ satisfy the properties in Problem 3.2 and suppose its gradient is Lipschitz continuous on the contractions $\mathcal{D}_\delta$ as in Property 3.3. Suppose further that $f$ is finite on the boundary of $\mathcal{D}$.*

*Then $f$ is uniformly continuous on $\mathcal{D}$ and has a modulus of continuity function $\omega$ quantifying its level of continuity, i.e. $|f(\boldsymbol{x}) - f(\boldsymbol{x}')| \leq \omega(\|\boldsymbol{x} - \boldsymbol{x}'\|) \ \forall \boldsymbol{x}, \boldsymbol{x}' \in \mathcal{D}$, with $\omega(\sigma) \downarrow 0$ as $\sigma \downarrow 0$.*

Figure 3: Illustration of the four points considered for the error analysis of the fixed-$\delta$ variant

*Let $\boldsymbol{x}^*$ be an optimal point of $f$ over $\mathcal{D}$. The iterates $\boldsymbol{x}^{(k)} \in \mathcal{D}_\delta$ of the FCFW algorithm as described in Algorithm 4 for a fixed $\delta > 0$ has sub-optimality over $\mathcal{D}$ bounded as:*

$$f(\boldsymbol{x}^{(k)}) - f(\boldsymbol{x}^*) \leq \frac{2\mathcal{C}_\delta}{(k+2)} + \omega\left(\delta \operatorname{diam}(\mathcal{D})\right), \tag{7}$$

*where $C_\delta \leq \operatorname{diam}(\mathcal{D}_\delta)^2 L_\delta$. Note that different norms can be used in the definition of $\omega(\cdot)$ and $C_\delta$.*

*Proof.* Let $\boldsymbol{x}^*_{(\delta)}$ be an optimal point of $f$ over $\mathcal{D}_\delta$. As $f$ has a Lipschitz continuous gradient over $\mathcal{D}_\delta$, we can use any standard convergence result of the Frank-Wolfe algorithm to bound the suboptimality of the iterate $\boldsymbol{x}^{(k)}$ over $\mathcal{D}_\delta$. Algorithm 4 (with a fixed $\delta$) describes the FCFW algorithm which guarantees at least as much progress as the standard FW algorithm (by step 15 and 20a), and thus we can use the convergence result from Jaggi (2013) as already stated in (4): $f(\boldsymbol{x}^{(k)}) - f(\boldsymbol{x}^*_{(\delta)}) \leq \frac{2C_\delta}{(k+2)}$ with $C_\delta \leq \operatorname{diam}(\mathcal{D}_\delta)^2 L_\delta$, where $L_\delta$ comes from Property 3.3. This gives the first term in (7). Note that if the function $f$ is *strongly* convex, then the FCFW algorithm has also a linear convergence rate (Lacoste-Julien and Jaggi, 2015), though we do not cover this here.

We now need to bound the difference $f(\boldsymbol{x}^*_{(\delta)}) - f(\boldsymbol{x}^*)$ coming from the fact that we are not optimizing over the full domain, and giving the second term in (7). We let $\tilde{\boldsymbol{x}}_{(\delta)}$ be the contraction of $\boldsymbol{x}^*$ on $\mathcal{D}_\delta$ towards $\boldsymbol{u}_0$, i.e. $\tilde{\boldsymbol{x}}_{(\delta)} := (1-\delta)\boldsymbol{x}^* + \delta\boldsymbol{u}_0$.[5] Note that $\|\tilde{\boldsymbol{x}}_{(\delta)} - \boldsymbol{x}^*\| = \delta\|\boldsymbol{x}^* - \boldsymbol{u}_0\| \leq \delta \operatorname{diam}(\mathcal{D})$, and thus can be made arbitrarily small by letting $\delta \downarrow 0$. Because $\tilde{\boldsymbol{x}}_{(\delta)} \in \mathcal{D}_\delta$, we have that $f(\tilde{\boldsymbol{x}}_{(\delta)}) \geq f(\boldsymbol{x}^*_{(\delta)})$ as $\boldsymbol{x}^*_{(\delta)}$ is optimal over $\mathcal{D}_\delta$. Thus $f(\boldsymbol{x}^*_{(\delta)}) - f(\boldsymbol{x}^*) \leq f(\tilde{\boldsymbol{x}}_{(\delta)}) - f(\boldsymbol{x}^*) \leq \omega(\|\tilde{\boldsymbol{x}}_{(\delta)} - \boldsymbol{x}^*\|)$ by the uniform continuity of $f$ (that we explain below). Since $\omega$ is an increasing function, we have $\omega(\|\tilde{\boldsymbol{x}}_{(\delta)} - \boldsymbol{x}^*\|) \leq \omega(\delta \operatorname{diam}(\mathcal{D}))$, giving us the control on the second term of (7). See Figure 3 for an illustration of the four points considered in this proof.

Finally, we explain why $f$ is uniformly continuous. As $f$ is a (lower semi-continuous) convex function, it is continuous at every point where it is finite. As $f$ is said to be finite at its boundary (and it is obviously finite in the relative interior of $\mathcal{D}$ as it is continuously differentiable there), then $f$ is continuous over the whole of $\mathcal{D}$. As $\mathcal{D}$ is compact, this means that $f$ is also uniformly continuous over $\mathcal{D}$. □

We note that the modulus of continuity function $\omega$ quantifies the level of continuity of $f$. For a Lipschitz continuous function, we have $\omega(\sigma) \leq L\sigma$. If instead we have $\omega(\sigma) \leq C\sigma^\alpha$ for some $\alpha \in [0,1]$, then $f$ is actually $\alpha$-Hölder continuous. We will see in Section E.2 that the TRW objective is not Lipschitz continuous, but it is $\alpha$-Hölder continuous for any $\alpha < 1$, and so is "almost" Lipschitz continuous. From the theorem, we see that to get an accuracy of the order $\epsilon$, we would need $(\delta \operatorname{diam}(\mathcal{D}))^\alpha < \epsilon$, and thus a contraction of $\delta < \frac{\epsilon^{(1/\alpha)}}{\operatorname{diam}(\mathcal{D})}$.

# D  Convergence with Adaptive-$\delta$

In this section, we show the convergence of the adaptive-$\delta$ FW algorithm to optimize a function $f$ satisfying the properties in Problem 3.2 and Property 3.3 (Lipschitz gradient over $\mathcal{D}_\delta$ with bounded growth).

The adaptive update for $\delta$ (given in Algorithm 1) can be used with the standard Frank-Wolfe optimization algorithm or also the fully corrective Frank-Wolfe (FCFW) variant. In FCFW, we ensure that every update makes more progress than a standard FW step with line-search, and thus we will show the convergence result in this section for standard FW (which also applies to FCFW). We describe the FCFW variant with approximate correction steps in Algorithm 4, as this is what we used in our experiments.

We first list a few definitions and lemmas that will be used for the main convergence convergence result given in Theorem D.6. We begin with the definitions of duality gaps that we use throughout this section. The Frank-Wolfe gap is our primary criterion for halting and measuring the progress of the optimization over $\mathcal{D}$. The uniform gap is a measure of the decrease obtainable from moving towards the uniform distribution.

**Definition D.1.** *We define the following gaps:*

1. *The Frank-Wolfe (FW) gap is defined as:* $g(\boldsymbol{x}^{(k)}) := \langle -\nabla f(\boldsymbol{x}^{(k)}), \boldsymbol{s}^{(k)} - \boldsymbol{x}^{(k)} \rangle.$

2. *The uniform gap is defined as:* $g_u(\boldsymbol{x}^{(k)}) := \langle -\nabla f(\boldsymbol{x}^{(k)}), \boldsymbol{u}_0 - \boldsymbol{x}^{(k)} \rangle.$

3. *The FW gap over $\mathcal{D}_\delta$ is:* $g_{(\delta^{(k)})}(\boldsymbol{x}^{(k)}) := \langle -\nabla f(\boldsymbol{x}^{(k)}), \boldsymbol{s}_{(\delta)}^{(k)} - \boldsymbol{x}^{(k)} \rangle.$

The name for the uniform gap comes from the fact that the FW gap over $\mathcal{D}_\delta$ can be expressed as a convex combination of the FW gap over $\mathcal{D}$ and the uniform gap:

$$
\begin{aligned}
g_{(\delta^{(k)})}(\boldsymbol{x}^{(k)}) &= \langle -\nabla f(\boldsymbol{x}^{(k)}),\ (1-\delta^{(k)})\boldsymbol{s}^{(k)} + \delta^{(k)}\boldsymbol{u}_0 - \boldsymbol{x}^{(k)} \rangle \\
&= (1-\delta^{(k)})g(\boldsymbol{x}^{(k)}) + \delta^{(k)}g_u(\boldsymbol{x}^{(k)}).
\end{aligned}
\tag{8}
$$

The uniform gap represents the negative directional derivative of $f$ at $\boldsymbol{x}^{(k)}$ in the direction $\boldsymbol{u}_0 - \boldsymbol{x}^{(k)}$. When the uniform gap is negative (thus $f$ is increasing when moving towards $\boldsymbol{u}_0$ from $\boldsymbol{x}^{(k)}$), then the contraction is hurting progress, which explains the type of adaptive update for $\delta$ given by Algorithm 1 where we consider shrinking $\delta$ in this case. This enables us to crucially relate the FW gap over $\mathcal{D}_\delta$ with the one over $\mathcal{D}$, as given in the following lemma, using the assumption that $\delta^{(\text{init})} \leq \frac{1}{4}$.

**Lemma D.2** (Gaps relationship). *For iterates progressing as in Algorithm 4 with adaptive update on $\delta$ as given in Algorithm 1, the gap over $\mathcal{D}_\delta$ and $\mathcal{D}$ are related as :* $g_{(\delta^{(k)})}(\boldsymbol{x}^{(k)}) \geq \frac{g(\boldsymbol{x}^{(k)})}{2}.$

*Proof.* The duality gaps $g(\boldsymbol{x}^{(k)})$ and $g_{(\delta^{(k)})}(\boldsymbol{x}^{(k)})$ computed as defined in (D.1) during Algorithm 4 are related by equation (8).

We analyze two cases separately:

(1) When $g_u(\boldsymbol{x}^{(k)}) \geq 0$, for $\delta^{(\text{init})} \leq \frac{1}{4}$, we have $g_{(\delta^{(k)})}(\boldsymbol{x}^{(k)}) \geq \frac{3}{4}g(\boldsymbol{x}^{(k)})$ as $\delta^{(k)} \leq \delta^{(\text{init})}$.

(2) When $g_u(\boldsymbol{x}^{(k)}) < 0$, from the update rule in lines 5 to 7 in Algorithm 1, we have $\delta^{(k)} \leq \frac{g(\boldsymbol{x}^{(k)})}{-4g_u(\boldsymbol{x}^{(k)})} \implies$ $\delta^{(k)}g_u(\boldsymbol{x}^{(k)}) \geq -\frac{g(\boldsymbol{x}^{(k)})}{4}$. Therefore, $g_{(\delta^{(k)})}(\boldsymbol{x}^{(k)}) \geq \frac{3}{4}g(\boldsymbol{x}^{(k)}) - \frac{g(\boldsymbol{x}^{(k)})}{4} = \frac{g(\boldsymbol{x}^{(k)})}{2}$.

Therefore, the gap over $\mathcal{D}_\delta$ and $\mathcal{D}$ are related as : $g_{(\delta^{(k)})}(\boldsymbol{x}^{(k)}) \geq \frac{g(\boldsymbol{x}^{(k)})}{2}$. $\qquad\square$

Another property that we will use in the convergence proof is that $-g_u$ is upper bounded for any convex function $f$:[6]

**Lemma D.3** (Bounded negative uniform gap). *Let $f$ be a continuously differentiable convex function on the relative interior of $\mathcal{D}$. Then for any fixed $\boldsymbol{u}_0$ in the relative interior of $\mathcal{D}$, $\exists B$ s.t.*

$$
\forall \boldsymbol{x} \in \mathcal{D},\ -g_u(\boldsymbol{x}) = \langle \nabla f(\boldsymbol{x}), \boldsymbol{u}_0 - \boldsymbol{x} \rangle \leq B.
\tag{9}
$$

*In particular, we can take the finite value:*

$$
B := \|\nabla f(\boldsymbol{u}_0)\|_* \mathrm{diam}_{\|\cdot\|}(\mathcal{D})
\tag{10}
$$

*Proof.* As $f$ is convex, its directional derivative is a monotone increasing function in any direction. Let $\boldsymbol{u}_0$ and $\boldsymbol{x}$ be points in the relative interior of $\mathcal{D}$; then their gradient exists and we have by the monotonicity property:

$$\langle \nabla f(\boldsymbol{u}_0) - \nabla f(\boldsymbol{x}), \boldsymbol{u}_0 - \boldsymbol{x} \rangle \geq 0$$
$$\implies \langle \nabla f(\boldsymbol{u}_0), \boldsymbol{u}_0 - \boldsymbol{x} \rangle \geq \langle \nabla f(\boldsymbol{x}), \boldsymbol{u}_0 - \boldsymbol{x} \rangle.$$

This inequality is valid for all $\boldsymbol{x}$ in the relative interior of $\mathcal{D}$, and can be extended to the boundary by taking limits (with potentially the RHS become minus infinity, but this is not a problem). Finally, by the definition of the dual norm (generalized Cauchy-Schwartz), we have $\langle \nabla f(\boldsymbol{u}_0), \boldsymbol{u}_0 - \boldsymbol{x} \rangle \leq \|\nabla f(\boldsymbol{u}_0)\|_* \|\boldsymbol{u}_0 - \boldsymbol{x}\| \leq \|\nabla f(\boldsymbol{u}_0)\|_* \mathrm{diam}_{\|\cdot\|}(\mathcal{D})$. $\qquad\square$

Finally, we need a last property of Algorithm 4 that allows us to bound the amount of perturbation $\delta^{(k)}$ of the polytope at every iteration as a function of the sub-optimality over $\mathcal{D}$.

**Lemma D.4** (Lower bound on perturbation). *Let $B$ be a bound such that $-8g_u(\boldsymbol{x}) \leq B$ for all $\boldsymbol{x} \in \mathcal{D}$ (given by Lemma D.3). Then at every stage of Algorithm 4, we have that:*

$$\delta^{(\mathrm{init})} \geq \delta^{(k)} \geq \min\left\{\frac{h_k}{B}, \delta^{(\mathrm{init})}\right\},$$

*where $\delta^{(\mathrm{init})}$ is the initial value of $\delta$ and $h_k := f(\boldsymbol{x}^{(k)}) - f(\boldsymbol{x}^*)$ is the sub-optimality of the iterate.*

*Proof.* When defining $\delta^{(k)}$ in step 10 of Algorithm 4, we either preserve the value of $\delta^{(k-1)}$ or if we update it, then by the lines 6 and 7 of Algorithm 1, we have $\delta^{(k)} \geq \frac{\tilde{\delta}}{2} = \frac{1}{2}\frac{g(\boldsymbol{x}^{(k)})}{-4g_u(\boldsymbol{x}^{(k)})} \geq \frac{g(\boldsymbol{x}^{(k)})}{B}$ (by using $g_u(\boldsymbol{x}^{(k)}) < 0$ in this case). Since $g(\boldsymbol{x}^{(k)}) \geq h_k$ (the FW gap always upper bounds the suboptimality), we conclude $\delta^{(k)} \geq \min\{\frac{h_k}{B}, \delta^{(k-1)}\}$. Unrolling this recurrence, we thus get:

$$\delta^{(k)} \geq \min\left\{\min_{0 \leq l \leq k} \frac{h_l}{B}, \delta^{(\mathrm{init})}\right\} = \min\left\{\frac{h_k}{B}, \delta^{(\mathrm{init})}\right\}.$$

For the last equality, we used the fact that $h_k$ is non-increasing since Algorithm 4 decreases the objective at every iteration (using the line-search in step 14). $\qquad\square$

We now bound the generalization of a standard recurrence that will arise in the proof of convergence. This is a generalization of the technique used in Teo et al. (2007) (also used in the context of Frank-Wolfe in the proof of Theorem C.4 in Lacoste-Julien et al. (2013)). The basic idea is that one can bound a recurrence inequality by the solution to a differential equation. We provide a detailed proof of the bound for completeness here.

**Lemma D.5** (Recurrence inequality solution). *Let $1 < a \leq b$. Suppose that $h_k$ is any non-negative sequence that satisfies the recurrence inequality:*

$$h_{k+1} \leq h_k - \frac{1}{bC_0}(h_k)^a \qquad \text{with initial condition} \quad h_0^{a-1} \leq C_0.$$

*Then $h_k$ is strictly decreasing (unless it equals zero) and can be bounded for $k \geq 0$ as:*

$$h_k \leq \left(\frac{C_0}{(\frac{a-1}{b})k + 1}\right)^{\frac{1}{a-1}}$$

*Proof.* Taking the continuous time analog of the recurrence inequality, we consider the differential equation:

$$\frac{\mathrm{d}h}{\mathrm{d}t} = \frac{-h^a}{bC_0} \qquad \text{with initial condition} \quad h(0) = C_0^{\frac{1}{a-1}}.$$

Solving it:

$$\frac{\mathrm{d}h}{\mathrm{d}t} = \frac{-h^a}{bC_0}$$

$$\implies \int \frac{\mathrm{d}h}{h^a} = \int \frac{-\mathrm{d}t}{bC_0}$$

$$\implies \left[\frac{-h^{1-a}}{a-1}\right]_{h(0)}^{h(t)} = -\frac{t-0}{bC_0}$$

*( Using the initial conditions:)*

$$\implies \frac{-1}{h(t)^{a-1}} + \frac{1}{C_0} = \frac{-t(a-1)}{bC_0}$$

$$\implies \frac{1}{h(t)^{a-1}} = \left((\frac{a-1}{b})t + 1\right)\frac{1}{C_0}$$

$$\implies h(t) = \left(\frac{C_0}{(\frac{a-1}{b})t+1}\right)^{\frac{1}{a-1}}.$$

We now denote the solution to the differential equation as $\tilde{h}(t)$. Note that it is a strictly decreasing convex function (which could also be directly implied from the differential equation as: $\frac{\mathrm{d}^2 h}{\mathrm{d}t^2} = -a \underbrace{\frac{h^{a-1}}{bC_0}}_{>0} \underbrace{h'(t)}_{<0} > 0$ ).

Our strategy will be to show by induction that if $h_k \leq \tilde{h}(k)$, then $h_{k+1} \leq \tilde{h}(k+1)$. This allows us to bound the recurrence by the solution to the differential equation.

Assume that $h_k \leq \tilde{h}(k)$. The base case is $h_0 \leq \tilde{h}(0) = C_0^{\frac{1}{a-1}}$, which is true by the initial condition on $h_0$.

Consider the utility function $l(h) := h - \frac{h^a}{bC_0}$ which is maximized at $\bar{h} := \left(\frac{bC_0}{a}\right)^{\frac{1}{a-1}}$. This function can be verified to be strictly concave for $a > 1$ and therefore is increasing for $h \leq \bar{h}$. Note that the recurrence inequality can be written as $h_{k+1} \leq l(h_k)$. Since $\tilde{h}$ is decreasing and that $\tilde{h}(0)) = C_0^{\frac{1}{a-1}} \leq \left(\frac{bC_0}{a}\right)^{\frac{1}{a-1}} = \bar{h}$ (the last inequality holds since $b \geq a$), we have $\tilde{h}(t) \leq \bar{h}$ for all $t \geq 0$, and so $\tilde{h}(t)$ is always in the monotone increasing region of $l$.

From the induction hypothesis and the monotonicity of $l$, we thus get that $l(h_k) \leq l(\tilde{h}(k))$.

Now the convexity of $\tilde{h}(t)$ gives us $\tilde{h}(k+1) \geq \tilde{h}(k) + \tilde{h}'(k) = \tilde{h}(k) - \frac{\tilde{h}(k)^a}{bC_0} = l(\tilde{h}(k))$. Combining these two facts with the recurrence inequality $h_{k+1} \leq l(h_k)$, we get: $h_{k+1} \leq l(h_k) \leq l(\tilde{h}(k)) \leq \tilde{h}(k+1)$, completing the induction step and the main part of the proof.

Finally, whenever $h_k > 0$, we have that $h_{k+1} < h_k$ from the recurrence inequality, and so $h_k$ is strictly decreasing as claimed. □

Given these elements, we are now ready to state the main convergence result for Algorithm 4. The convergence rate goes through three stages with increasingly slower rate. The level of suboptimality $h_k$ determines the stage. We first give the high level intuition behind these stages. Recall that by Lemma D.4, $h_k$ lower bounds the amount of perturbation $\delta^{(k)}$, and thus when $h_k$ is big, the function $f$ is well-behaved by Property 3.3. In the first stage, the suboptimality is bigger than some target constant (which implies that the FW gap is big), yielding a geometric rate of decrease of error (as is standard for FW with line-search in the first few steps). In the second stage, the suboptimality is in an intermediate regime: it is smaller than the target constant, but big enough compared to the initial $\delta^{\text{init}}$ so that $f$ is still well-behaved on $\mathcal{D}_{\delta^{(k)}}$. We get there the usual $O(1/k)$ rate as in standard FW. Finally, in the third stage, we get the slower $O(k^{-\frac{1}{p+1}})$ rate where the growth in $O(\delta^{-p})$ of the Lipschitz constant of $f$ over $\mathcal{D}_\delta$ comes into play.

**Theorem D.6** (Global convergence for adaptive-$\delta$ variant over $\mathcal{D}$). *Consider the optimization of $f$ satisfying the properties in Problem 3.2 and Property 3.3. Let $\widetilde{C} := L \operatorname{diam}_{\|\cdot\|}(\mathcal{D})^2$, where $L$ is from Property 3.3. Let $B$ be the upper bound on the negative uniform gap: $-8g_u(\boldsymbol{x}) \leq B$ for all $\boldsymbol{x} \in \mathcal{D}$, as used in Lemma D.4 (arising from Lemma D.3). Then the iterates $\boldsymbol{x}^{(k)}$ obtained by running the Frank-Wolfe updates over $\mathcal{D}_\delta$ with line-search with $\delta$ updated according to Algorithm 1 (or as summarized in a FCFW variant in Algorithm 4), have suboptimality $h_k$ upper bounded as:*

1. $h_k \leq \left(\frac{1}{2}\right)^k h_0 + \frac{\widetilde{C}}{\delta_0^p}$ *for $k$ such that $h_k \geq \max\{B\delta_0, \frac{2\widetilde{C}}{\delta_0^p}\}$,*

2. $h_k \le \frac{2\widetilde{C}}{\delta_0^p} \left[ \frac{1}{\frac{1}{4}(k-k_0)+1} \right]$ *for k such that* $B\delta_0 \le h_k \le \frac{2\widetilde{C}}{\delta_0^p}$,

3. $h_k \le \left[ \frac{\max(\widetilde{C}, B\delta_0^{p+1})B^p}{\frac{p+1}{\max(8,p+2)}(k-k_1)+1} \right]^{\frac{1}{p+1}} = O(k^{-\frac{1}{p+1}})$ *for k such that* $h_k \le B\delta_0$,

*where* $\delta_0 = \delta^{(\mathrm{init})}$, $h_0$ *is the initial suboptimality, and* $k_0$ *and* $k_1$ *are the number of steps to reach stage 2 and 3 respectively which are bounded as:* $k_0 \le \max(0, \lceil \log_{\frac{1}{2}} \frac{\widetilde{C}}{h_0 \delta_0^p} \rceil)$, $k_0 \le k_1 \le k_0 + \max\left(0, \lceil \frac{8\widetilde{C}}{B\delta_0^{p+1}} \rceil - 4\right)$.

*Proof.* Let $\boldsymbol{x}_\gamma := \boldsymbol{x}^{(k)} + \gamma \boldsymbol{d}_k^{\mathrm{FW}}$ with $\boldsymbol{d}_k^{\mathrm{FW}}$ defined in step 12 in Algorithm 4. Note that $\boldsymbol{x}_\gamma \in \mathcal{D}_\delta$ with $\delta = \delta^{(k)}$ for all $\gamma \in [0,1]$. We apply the Descent Lemma A.1 on this update to get:

$$f(\boldsymbol{x}_\gamma) \le f(\boldsymbol{x}^{(k)}) + \gamma \langle \nabla f(\boldsymbol{x}^{(k)}), \boldsymbol{d}_k^{\mathrm{FW}} \rangle + \gamma^2 \frac{L\|\boldsymbol{d}_k^{\mathrm{FW}}\|^2}{2(\delta^{(k)})^p} \qquad \forall \gamma \in [0,1].$$

We have $L\|\boldsymbol{d}_k^{\mathrm{FW}}\|^2 \le \widetilde{C}$ by assumption and $\langle \nabla f(\boldsymbol{x}^{(k)}), \boldsymbol{d}_k^{\mathrm{FW}} \rangle = -g_{(\delta)}(\boldsymbol{x}^{(k)})$ by definition. Moreover, $\boldsymbol{x}^{(k+1)}$ is defined to make at least as much progress than the line-search result $\min_{\gamma \in [0,1]} f(\boldsymbol{x}_\gamma)$ (line 14 and 15), and so we have:

$$f(\boldsymbol{x}^{(k+1)}) \le f(\boldsymbol{x}^{(k)}) - \gamma g_{(\delta)}(\boldsymbol{x}^{(k)}) + \gamma^2 \frac{\widetilde{C}}{2(\delta^{(k)})^p} \quad \forall \gamma \in [0,1]$$

$$\le f(\boldsymbol{x}^{(k)}) - \frac{\gamma}{2} g(\boldsymbol{x}^{(k)}) + \gamma^2 \frac{\widetilde{C}}{2(\delta^{(k)})^p} \quad \forall \gamma \in [0,1].$$

For the final inequality, we used Lemma D.2 which relates the gap over $\mathcal{D}_\delta$ to the gap over $\mathcal{D}$.

Subtracting $f(\boldsymbol{x}^*)$ from both sides and using $g(\boldsymbol{x}^{(k)}) \ge h_k$ by convexity, we get:

$$h_{k+1} \le h_k - \frac{\gamma h_k}{2} + \frac{\gamma^2 \widetilde{C}}{2(\delta^{(k)})^p}.$$

Now, using Lemma D.4, we have that $\delta^{(k)} \ge \min(\frac{h_k}{B}, \delta^{(\mathrm{init})})$:

$$h_{k+1} \le h_k - \gamma \frac{h_k}{2} + \frac{\gamma^2}{2} \frac{\widetilde{C}}{\left(\min(\frac{h_k}{B}, \delta^{(\mathrm{init})})\right)^p} \quad \forall \gamma \in [0,1]. \tag{11}$$

We refer to (11) as the master inequality. Since we no longer have a dependance on $\delta^{(k)}$, we refer to $\delta^{(\mathrm{init})}$ as $\delta_0$. We now follow a similar form of analysis as in the proof of Theorem C.4 in Lacoste-Julien et al. (2013). To solve this and bound the suboptimality, we consider three stages:

1. **Stage 1:** The min in the denominator is $\delta_0$ and $h_k$ is big: $h_k \ge \max\{B\delta_0, \frac{2\widetilde{C}}{\delta_0^p}\}$.

2. **Stage 2:** The min in the denominator is $\delta_0$ and $h_k$ is small: $B\delta_0 \le h_k \le \frac{2\widetilde{C}}{\delta_0^p}$.

3. **Stage 3:** The min in the denominator is $\frac{h_k}{B}$, i.e.: $h_k \le B\delta_0$.

Since $h_k$ is decreasing, once we leave a stage, we no longer re-enter it. The overall strategy for each stage is as follows. For each recurrence that we get, we select a $\gamma^*$ that realizes the tightest upper bound on it.

Since we are restricted that $\gamma^* \in [0,1]$, we have to consider when $\gamma^* > 1$ and $\gamma^* \le 1$. For the former, we bound the recurrence obtained by substituting $\gamma = 1$ into (11). For the latter, we substitute the form of $\gamma^*$ into the recurrence and bound the result.

**Stage 1**

We consider the case where $h_k \ge B\delta_0$. This yields:

$$h_{k+1} \le h_k - \frac{\gamma h_k}{2} + \frac{\gamma^2 \widetilde{C}}{2(\delta_0)^p} \tag{12}$$

The bound is minimized by setting $\gamma^* = \frac{h_k \delta_0^p}{2\widetilde{C}}$. On the other hand, the bound is only valid for $\gamma \in [0, 1]$, and thus if $\gamma^* > 1$, i.e. $h_k > \frac{2\widetilde{C}}{\delta_0^p}$ (stage 1), then $\gamma = 1$ will yield the minimum feasible value for the bound. Unrolling the recursion (12) for $\gamma = 1$ during this stage (where $h_l > \frac{2\widetilde{C}}{\delta_0^p}$ for $l < k$ as $h_k$ is decreasing), we get:

$$
\begin{aligned}
h_{k+1} &\leq \frac{h_k}{2} + \frac{\widetilde{C}}{2\delta_0^p} \\
&\leq \frac{1}{2}\left(\frac{h_{k-1}}{2} + \frac{\widetilde{C}}{2\delta_0^p}\right) + \frac{\widetilde{C}}{2\delta_0^p} \\
&\leq \left(\frac{1}{2}\right)^{k+1} h_0 + \frac{\widetilde{C}}{2\delta_0^p} \underbrace{\sum_{l=0}^{k}\left(\frac{1}{2}\right)^l}_{\leq \sum_{l=0}^{\infty}\left(\frac{1}{2}\right)^l = 2}
\end{aligned}
$$

$$
\text{thus} \quad h_k \leq \left(\frac{1}{2}\right)^k h_0 + \frac{\widetilde{C}}{\delta_0^p}, \tag{13}
$$

giving the bound for the iterates in the first stage.

We can compute an upper bound on the number of steps it takes to reach a suboptimality of $\frac{2\widetilde{C}}{\delta_0^p}$ by looking at the minimum $k$ which ensures that the bound in (13) becomes smaller than $\frac{2\widetilde{C}}{\delta_0^p}$, yielding $k_{\max} = \max(0, \lceil \log_{\frac{1}{2}} \frac{\widetilde{C}}{h_0 \delta_0^p} \rceil)$. Therefore, let $k_0 \leq k_{\max}$ be the first $k$ such that $h_k \leq \frac{2\widetilde{C}}{\delta_0^p}$.

## Stage 2

For this case analysis, we refer to $k$ as being the iterations *after* $k_0$ steps have elapsed. I.e. if $k_{\text{new}} := k - k_0$, then we refer to $k_{\text{new}}$ as $k$ moving forward.

In stage 2, we suppose that $B\delta_0 \leq h_k \leq \frac{2\widetilde{C}}{\delta_0^p}$. This means that $\gamma^* = \frac{h_k \delta_0^p}{2\widetilde{C}} \leq 1$.

Substituting $\gamma = \gamma^*$ into (12) yields: $h_{k+1} \leq h_k - h_k^2 \frac{\delta_0^p}{8\widetilde{C}}$.

Using the result of Lemma D.5 with $a = 2$, $b = 4$ and $C_0 = \frac{2\widetilde{C}}{\delta_0^p}$, we get the bound:

$$
h_k \leq \frac{\frac{2\widetilde{C}}{\delta_0^p}}{\frac{k-k_0}{4} + 1}.
$$

It is worthwhile to point out at this juncture that the bound obtained for stage 2 is the same as the one for regular Frank-Wolfe, but with a factor of 4 worse due to the factor of $\frac{1}{2}$ in front of the FW gap which appeared due to Lemma D.2.

## Stage 3

Here, we suppose $h_k \leq B\delta_0$. We can compute a bound on the number of steps $k_1$ needed get to stage 3 by looking at the number of steps it takes for the bound in stage 2 to becomes less than $B\delta_0$:

$$
\begin{aligned}
\frac{2\widetilde{C}}{\delta_0^p}\left[\frac{4}{k_1 - k_0 + 4}\right] &\leq B\delta_0 \\
\left[\frac{1}{k_1 - k_0 + 4}\right] &\leq \frac{B\delta_0^{p+1}}{8\widetilde{C}} \\
k_1 &\geq k_0 + \lceil \frac{8\widetilde{C}}{B\delta_0^{p+1}} \rceil - 4.
\end{aligned}
$$

As before, moving forward, our notation on $k$ represents the number of steps taken after $k_1$ steps.

Then, the master inequality (11) becomes:

$$h_{k+1} \le h_k - \frac{\gamma}{2}h_k + \frac{\gamma^2 \widetilde{C} B^p}{2h_k^p}.$$

To simplify the rest of the analysis, we replace $\widetilde{C}B^p$ with $F := \max(B\delta_0^{p+1}, \widetilde{C})B^p$. We then get the bound:

$$h_{k+1} \le h_k - \frac{\gamma}{2}h_k + \frac{\gamma^2 F}{2h_k}, \tag{14}$$

which is minimized by setting $\gamma^* := \frac{h_k^{p+1}}{2F}$. Since $F \ge B^{p+1}\delta_0^{p+1}$ (by construction) and $h_k^{p+1} \le (B\delta_0)^{p+1}$ (by the condition to be in stage 3), we necessarily have that $\gamma^* \le 1$. We chose the value of $F$ to avoid having to consider the possibility $\gamma^* > 1$ as we did in the distinction between stage 1 and stage 2.

Hence, substituting $\gamma = \gamma^*$ in (14), we get:

$$h_{k+1} \le h_k - \frac{h_k^{p+2}}{8F}.$$

Using the result of Lemma D.5 with $a = p+2$, $b = \max(8, p+2)$ and $C_0 = F$, we get the bound:

$$h_k \le \left[ \frac{\max(\widetilde{C}, B\delta_0^{p+1})B^p}{\frac{p+1}{\max(8, p+2)}(k - k_1) + 1} \right]^{\frac{1}{p+1}} = O(k^{-\frac{1}{p+1}}),$$

concluding the proof. $\qquad\qquad\qquad\qquad\qquad\qquad\qquad\qquad\qquad\qquad\qquad\qquad\qquad\qquad\square$

Interestingly, the obtained rate of $O(1/\sqrt{k})$ for $p = 1$ (for the TRW objective e.g.) is the standard rate that one would get for the optimization of a general non-smooth convex function with the projected subgradient method (and it is even a lower bound for some class of first-order methods; see e.g. Section 3.2 in Nesterov (2004)). The fact that our function $f$ does not have Lipschitz continuous gradient on the whole domain brings us back to the realm of non-smooth optimization. It is an open question whether Algorithm 4 has an optimal rate for the class of functions defined in the assumptions of Theorem D.6.

# E  Properties of the TRW Objective

In this section, we explicitly compute bounds for the constants appearing in the convergence statements for our fixed-$\delta$ and adaptive-$\delta$ algorithms for the optimization problem given by:

$$\min_{\vec{\mu} \in \mathcal{M}} -\text{TRW}(\vec{\mu}; \vec{\theta}, \boldsymbol{\rho}).$$

In particular, we compute the Lipschitz constant for its gradient over $\mathcal{M}_\delta$ (Property 3.3), we give a form for its modulus of continuity function $\omega(\cdot)$ (used in Theorem 3.4), and we compute $B$, the upper bound on the negative uniform gap (as used in Lemma D.3).

## E.1  Property 3.3 : Controlled Growth of Lipschitz Constant over $\mathcal{M}_\delta$

We first motivate our choice of norm over $\mathcal{M}$. Recall that $\vec{\mu}$ can be decomposed into $|V|+|E|$ blocks, with one pseudo-marginal vector $\boldsymbol{\mu}_i \in \Delta_{\text{VAL}_i}$ for each node $i \in V$, and one vector $\boldsymbol{\mu}_{ij} \in \Delta_{\text{VAL}_i \text{VAL}_j}$ per edge $\{i, j\} \in E$, where $\Delta_d$ is the probability simplex over $d$ values. We let $c$ be the cliques in the graph (either nodes or edges). From its definition in (2), $f(\vec{\mu}) := -\text{TRW}(\vec{\mu}; \vec{\theta}, \boldsymbol{\rho})$ decomposes as a separable sum of functions of each block only:

$$f(\vec{\mu}) := -\text{TRW}(\vec{\mu}; \vec{\theta}, \boldsymbol{\rho}) = -\sum_c (K_c H(\boldsymbol{\mu}_c) + \langle \boldsymbol{\theta}_c, \boldsymbol{\mu}_c \rangle) =: \sum_c g_c(\boldsymbol{\mu}_c), \tag{15}$$

where $K_c$ is $(1 - \sum_{j \in \mathcal{N}(i)} \rho_{ij})$ if $c = i$ and $\rho_{ij}$ if $c = \{i, j\}$. The function $g_c$ also decomposes as a separable sum:

$$g_c(\boldsymbol{\mu}_c) := \sum_{x_c} K_c \mu_c(x_c) \log(\mu_c(x_c)) - \theta_c(x_c)\mu_c(x_c) =: \sum_{x_c} g_{c,x_c}(\mu_c(x_c)). \tag{16}$$

As $\mathcal{M}$ is included in a product of probability simplices, we will use the natural $\ell_\infty/\ell_1$ block-norm, i.e. $\|\vec{\mu}\|_{\infty,1} := \max_c \|\boldsymbol{\mu}_c\|_1$. The diameter of $\mathcal{M}$ in this norm is particularly small: $\text{diam}_{\|\cdot\|_{\infty,1}}(\mathcal{M}) \le 2$. The dual norm of the $\ell_\infty/\ell_1$ block-norm is the $\ell_1/\ell_\infty$ block-norm, which is what we will need to measure the Lipschitz constant of the gradient (because of the dual norm pairing requirement from the Descent Lemma A.1).

**Lemma E.1.** *Consider the $\ell_\infty/\ell_1$ norm on $\mathcal{M}$ and its dual norm $\ell_1/\ell_\infty$ to measure the gradient. Then $\nabla TRW(\vec{\mu}; \vec{\theta}, \rho)$ is Lipschitz continuous over $\mathcal{M}_\delta$ with respect to these norms with Lipschitz constant $L_\delta \leq \frac{L}{\delta}$ with:*

$$L \leq 4|V| \max_{ij \in E} (VAL_i VAL_j). \tag{17}$$

*Proof.* We first consider one scalar component of the separable $g_c(\boldsymbol{\mu}_c)$ function given in (16) (i.e. for one $\mu_c(x_c)$ coordinate). Its derivative is $K_c(1 + \log(\mu_c(x_c)) - \theta_c(x_c)$ with second derivative $\frac{K_c}{\mu_c(x_c)}$. If $\vec{\mu} \in \mathcal{M}_\delta$, then we have $\mu_c(x_c) \geq \delta u_0(x_c) = \frac{\delta}{n_c}$, where $n_c$ is the number of possible values that the assignment variable $x_c$ can take. Thus for $\vec{\mu} \in \mathcal{M}_\delta$, we have that the $x_c$-component of $g_c$ is Lipschitz continuous with constant $|K_c|n_c/\delta$. We thus have:

$$\|\nabla g_c(\boldsymbol{\mu}_c) - \nabla g_c(\boldsymbol{\mu}'_c)\|_\infty = \max_{x_c} |g'_{c,x_c}(\mu(x_c)) - g'_{c,x_c}(\mu'(x_c))|$$

$$\leq \frac{|K_c|n_c}{\delta} \|\boldsymbol{\mu}_c - \boldsymbol{\mu}'_c\|_\infty \leq \frac{|K_c|n_c}{\delta} \|\boldsymbol{\mu}_c - \boldsymbol{\mu}'_c\|_1.$$

Considering now the $\ell_1$-sum over blocks, we have:

$$\|\nabla f(\vec{\mu}) - \nabla f(\vec{\mu}')\|_{1,\infty} = \sum_c \|\nabla g_c(\boldsymbol{\mu}_c) - \nabla g_c(\boldsymbol{\mu}'_c)\|_\infty$$

$$\leq \sum_c \frac{K_c n_c}{\delta} \|\boldsymbol{\mu}_c - \boldsymbol{\mu}'_c\|_1 \leq \frac{1}{\delta} \left(\sum_c K_c n_c\right) \|\vec{\mu} - \vec{\mu}'\|_{\infty,1}.$$

The Lipschitz constant is thus indeed $\frac{L}{\delta}$ with $L := \sum_c |K_c| n_c$. Let us first consider the sum for $c \in V$; we have $K_i = 1 - \sum_{j \in \mathcal{N}(i)} \rho_{ij}$. Thus:

$$\sum_i |K_i| \leq |V| + \sum_i \sum_{j \in \mathcal{N}(i)} \rho_{ij}$$

$$= |V| + 2 \sum_{ij \in E} \rho_{ij} = |V| + 2(|V| - 1) \leq 3|V|.$$

Here we used the fact that $\rho_{ij}$ came from the marginal probability of edges of spanning trees (and so with $|V| - 1$ edges). Similarly, we have $\sum_{ij \in E} |K_{ij}| \leq |V|$. Combining these we get:

$$L = \sum_c |K_c| n_c \leq (\max_c n_c) \sum_c |K_c| \leq \max_{ij \in E} VAL_i VAL_j 4|V|. \tag{18}$$

$\square$

*Remark* 1. The important quantity in the convergence of Frank-Wolfe type algorithms is $\tilde{C} = L \operatorname{diam}(\mathcal{M})^2$. We are free to take any dual norm pairs to compute this quantity, but some norms are better aligned with the problem than others. Our choice of norm in Lemma E.1 gives $\tilde{C} \leq 16|V|k^2$ where $k$ is the maximum number of possible values a random variable can take. It is interesting that $|E|$ does not appear in the constant. If instead we had used the $\ell_2/\ell_1$ block-norm on $\mathcal{M}$, we get that $\operatorname{diam}_{\ell_2/\ell_1}(\mathcal{M})^2 = 4(|V| + |E|)$, while the constant $L$ with dual norm $\ell_2/\ell_\infty$ would be instead $\max_c |K_c| n_c$ which is bigger than $\max_c n_c = k^2$, thus giving a worse bound.

## E.2 Modulus of Continuity Function

We begin by computing a modulus of continuity function for $-x \log x$ with an additive linear term.

**Lemma E.2.** *Let $g(x) := -Kx \log x + \theta x$. Consider $x, x' \in [0, 1]$ such that $|x - x'| \leq \sigma$, then:*

$$|g(x') - g(x)| \leq \sigma|\theta| + 2\sigma|K| \max\{-\log(2\sigma), 1\} =: \omega_g(\sigma). \tag{19}$$

*Proof.* Without loss of generality assume $x' > x$, then we have two cases:

**Case i.** If $x > \sigma$, then we have that the Lipschitz constant of $g(x)$ is $L_\sigma = |\theta| + |K||(1 + \log \sigma)|$ (obtained by taking the supremum of its derivative). Therefore, we have that $|g(x') - g(x)| \leq L_\sigma \sigma$. Note that $L_\sigma \sigma \to 0$ when $\sigma \to 0$ even if $L_\sigma \to \infty$, since $L_\sigma$ grows logarithmically.

**Case ii.** If $x \leq \sigma$, then $x' \leq x + \sigma \leq 2\sigma$. Therefore:

$$|g(x') - g(x)| \leq |K||x \log x - x' \log x'| + |\theta||x' - x|. \tag{20}$$

Now, we have that $-x \log x$ is non-negative for $x \in [0, 1]$. Furthermore, we have that $-x \log x$ is increasing when $x < \exp(-1)$ and decreasing afterwards. First suppose that $2\sigma \leq \exp(-1)$; then $-x' \log x' \geq -x \log x \geq 0$ which implies:

$$|x \log x - x' \log x'| \leq -x' \log x' \leq -2\sigma \log(2\sigma).$$

In the case $2\sigma > \exp(-1)$, then we have:

$$|x \log x - x' \log x'| \leq \max_{y \in [0,1]} \{-y \log y\} = \exp(-1) \leq 2\sigma.$$

Combining these two possibilities, we get:

$$|x \log x - x' \log x'| \leq 2\sigma \max\{-\log(2\sigma), 1\}.$$

The inequality (20) thus becomes:

$$|g(x') - g(x)| \leq |K| 2\sigma \max\{-\log(2\sigma), 1\} + |\theta|\sigma,$$

which is what we wanted to prove. $\qquad\square$

For small $\sigma$, the dominant term of the function $\omega_g(\sigma)$ in Lemma E.2 is of the form $C \cdot -\sigma \log \sigma$ for a constant $C$. If we require that this be smaller than some small $\xi > 0$, then we can choose an approximate $\sigma$ by solving for $x$ in $-Ax \log x = \xi$ yielding $x = \exp(W_{-1} \frac{\xi}{A})$ where $W_{-1}$ is the negative branch of the Lambert W-function. This is almost linear and yields approximately $x = O(\xi)$ for small $\xi$. In fact, we have that $\omega_g(\sigma) \leq C'\sigma^\alpha$ for any $\alpha < 1$, and thus $g$ is "almost" Lipschitz continuous.

**Lemma E.3.** *The following function is a modulus of continuity function for the* $TRW(\vec{\mu}; \vec{\theta}, \rho)$ *objective over* $\mathcal{M}$ *with respect to the $\ell_\infty$ norm:*

$$\omega(\sigma) := \sigma\|\theta\|_1 + 2\sigma\tilde{K}\max\{-\log(2\sigma), 1\}, \tag{21}$$

*where* $\tilde{K} := 4|V|\max_{ij \in E} VAL_i VAL_j$.

*That is, for* $\vec{\mu}, \vec{\mu}' \in \mathcal{M}$ *with* $\|\vec{\mu}' - \vec{\mu}\|_\infty \leq \sigma$, *we have:*

$$|TRW(\vec{\mu}; \vec{\theta}, \rho) - TRW(\vec{\mu}'; \vec{\theta}, \rho)| \leq \omega(\sigma).$$

*Proof.* $\text{TRW}(\vec{\mu}; \vec{\theta}, \rho)$ can be decomposed into functions of the form $-Kx \log x + \theta x$ (see (15) and (16)) and so we apply the Lemma E.2 element-wise. Let $c$ index the clique component in the marginal vector.

$$
\begin{aligned}
|\text{TRW}(\vec{\mu}; \vec{\theta}, \rho) - \text{TRW}(\vec{\mu}'; \vec{\theta}, \rho)| &= \sum_c \sum_{x_c} |g_{c,x_c}(\mu_c(x_c)) - g_{c,x_c}(\mu'_c(x_c))| \\
&\quad (\textit{Using Lemma E.2 and } \|\vec{\mu}' - \vec{\mu}\|_\infty \leq \sigma) \\
&\leq \sum_c \sum_{x_c} (|K_c| 2\sigma \max\{-\log(2\sigma), 1\} + |\theta(x_c)|\sigma) \\
&= 2\sigma \max\{-\log(2\sigma), 1\} \sum_c |K_c| n_c + \|\theta\|_1 \sigma,
\end{aligned}
$$

where we recall $n_c$ is the number of values that $x_c$ can take. By re-using the bound on $\sum_c |K_c| n_c$ from (18), we get the result. $\qquad\square$

## E.3 Bounded Negative Uniform Gap

**Lemma E.4** (Bound for the negative uniform gap of TRW objective)**.** *For the negative TRW objective* $f(\vec{\mu}) := -TRW(\vec{\mu}; \vec{\theta}, \rho)$, *the bound $B$ on the negative uniform gap as given in Lemma D.3 for $\mathbf{u}_0$ being the uniform distribution can be taken as:*

$$B = 2\sum_c \max_{x_c} |\theta_c(x_c)| =: 2\|\vec{\theta}\|_{1,\infty} \tag{22}$$

*Proof.* From Lemma D.3, we want to bound $\|\nabla f(\mathbf{u}_0)\|_* = \|\vec{\theta} + \nabla_{\vec{\mu}} H(\mathbf{u}_0; \rho))\|_*$. The clique entropy terms $H(\mu_c)$ are maximized by the uniform distribution, and thus $\mathbf{u}_0$ is a stationary point of the TRW entropy function with zero gradient. We can thus simply take $B = \|\vec{\theta}\|_* \text{diam}_{\|\cdot\|}(\mathcal{M})$. By taking the $\ell_\infty/\ell_1$ norm on $\mathcal{M}$, we get a diameter of 2, giving the given bound. $\qquad\square$

## E.4 Summary

We now give the details of suboptimality guarantees for our suggested algorithm to optimize $f(\vec{\mu}) := -\text{TRW}(\vec{\mu}; \vec{\theta}, \rho)$ over $\mathcal{M}$. The (strong) convexity of the negative TRW objective is shown in (Wainwright et al., 2005; London et al., 2015). $\mathcal{M}$ is the convex hull of a finite number of vectors representing assignments to random variables and therefore a compact convex set. The entropy function is continously differentiable on the relative interior of the probability simplex, and thus the TRW objective has the same property on the relative interior of $\mathcal{M}$. Thus $-\text{TRW}(\vec{\mu}; \vec{\theta}, \rho)$ satisfies the properties laid out in Problem 3.2.

**Lemma E.5** (Suboptimality bound for optimizing $-\text{TRW}(\vec{\mu}; \vec{\theta}, \rho)$ with the fixed-$\delta$ algorithm). *For the optimization of $-\text{TRW}(\vec{\mu}; \vec{\theta}, \rho)$ over $\mathcal{M}_\delta$ with $\delta \in (0, 1]$, the suboptimality is bounded as:*

$$TRW(\vec{\mu}^*; \vec{\theta}, \rho) - TRW(\vec{\mu}^{(k)}; \vec{\theta}, \rho) \leq \frac{2\mathcal{C}_\delta}{(k+2)} + \omega(2\delta), \tag{23}$$

*with $\vec{\mu}^*$ the optimizer of $TRW(\vec{\mu}; \vec{\theta}, \rho)$ in $\mathcal{M}$, where $C_\delta \leq 16\frac{|V|\max_{(ij)\in E} VAL_i VAL_j}{\delta}$, and $\omega(\sigma) = \sigma\|\vec{\theta}\|_1 + 2\sigma\tilde{K}\max\{-\log(2\sigma), 1\}$, where $\tilde{K} := 4|V|\max_{ij\in E} VAL_i VAL_j$.*

*Proof.* Using $\text{diam}_{\|\cdot\|_{\infty,1}}(\mathcal{M}) \leq 2$, and $L_\delta$ from Lemma E.1, we can compute $C_\delta \leq \text{diam}(\mathcal{M})^2 L_\delta$. Lemma E.3 computes the modulus of continuity $\omega(\sigma)$. The rate then follows directly from Theorem C.1. □

**Lemma E.6** (Global convergence rate for optimizing $-\text{TRW}(\vec{\mu}; \vec{\theta}, \rho)$ with the adaptive-$\delta$ algorithm). *Consider the optimization of $-\text{TRW}(\vec{\mu}; \vec{\theta}, \rho)$ over $\mathcal{M}$ with the optimum given by $\vec{\mu}^*$. The iterates $\vec{\mu}^{(k)}$ obtained by running the Frank-Wolfe updates over $\mathcal{M}_\delta$ using line-search with $\delta$ updated according to Algorithm 1 (or as summarized in a FCFW variant in Algorithm 4), have suboptimality $h_k = TRW(\vec{\mu}^*; \vec{\theta}, \rho) - TRW(\vec{\mu}^{(k)}; \vec{\theta}, \rho)$ upper bounded as:*

*1. $h_k \leq \left(\frac{1}{2}\right)^k h_0 + \frac{\widetilde{C}}{\delta_0}$ for k such that $h_k \geq \max\{B\delta_0, \frac{2\widetilde{C}}{\delta_0}\}$,*

*2. $h_k \leq \frac{2\widetilde{C}}{\delta_0}\left[\frac{1}{\frac{1}{4}(k-k_0)+1}\right]$ for k such that $B\delta_0 \leq h_k \leq \frac{2\widetilde{C}}{\delta_0}$,*

*3. $h_k \leq \left[\frac{\max(\widetilde{C}, B\delta_0^2)B}{\frac{1}{4}(k-k_1)+1}\right]^{\frac{1}{2}} = O(k^{-\frac{1}{2}})$ for k such that $h_k \leq B\delta_0$,*

*where*

- $\delta_0 = \delta^{(\text{init})} \leq \frac{1}{4}$

- $\widetilde{C} := 16|V|\max_{(ij)\in E}(VAL_i VAL_j)$

- $B = 16\|\vec{\theta}\|_{1,\infty}$

- $h_0$ *is the initial suboptimality*

- $k_0$ *and $k_1$ are the number of steps to reach stage 2 and 3 respectively which are bounded as:* $k_0 \leq \max(0, \lceil\log_{\frac{1}{2}}\frac{\widetilde{C}}{h_0\delta_0}\rceil)$ $k_0 \leq k_1 \leq k_0 + \max\left(0, \lceil\frac{8\widetilde{C}}{B\delta_0^2}\rceil - 4\right)$

*Proof.* Using $\text{diam}_{\|\cdot\|_{\infty,1}}(\mathcal{M}) \leq 2$, we bound $\widetilde{C} \leq L\,\text{diam}_{\|\cdot\|_{\infty,1}}(\mathcal{M})^2$ with $L$ (from Property 3.3) derived in Lemma E.1. We bound $-8g_u(\vec{\mu}^{(k)})$ (the upper bound on the negative uniform gap) using the value derived in Lemma E.4. The rate then follows directly from Theorem D.6 using $p = 1$ (see Lemma E.1 where $L_\delta \leq \frac{L}{\delta}$). □

The dominant term in Lemma E.6 is $\widetilde{C}B\,k^{-\frac{1}{2}}$, with $\widetilde{C}B = O(\|\vec{\theta}\|_{1,\infty}|V|)$. We thus find that both bounds depend on norms of $\vec{\theta}$. This is unsurprising since large potentials drive the solution of the marginal inference problem away from the centre of $\mathcal{M}$, corresponding to regions of high entropy, and towards the boundary of the polytope (lower entropy). Regions of low entropy correspond to smaller components of the marginal vector,

which in turn result in larger and poorly behaved gradients of $-\text{TRW}(\vec{\boldsymbol{\mu}}; \vec{\boldsymbol{\theta}}, \boldsymbol{\rho})$, which slows down the resulting optimization.

# F  Correction and Local Search Steps in Algorithm 2

Algorithm 5 details the **CORRECTION** procedure used in line 16 of Algorithm 2 to implement the correction step of the FCFW algorithm. It uses the modified Frank-Wolfe algorithm (FW with away steps), as detailed in Algorithm 3. Algorithm 6 depicts the **LOCALSEARCH** procedure used in line 17 of Algorithm 2. The local search is performing FW over $\mathcal{M}_\delta$ for a fixed $\delta$ using the iterated conditional mode algorithm as an approximate FW oracle. This enables the finding in a cheap of way of more vertices to augment the correction polytope $V$.

---

**Algorithm 5:** Re-Optimizing over correction polytope $V$ using MFW, $f$ is the negative TRW objective

1: **CORRECTION**$(\boldsymbol{x}^{(0)}, V, \delta, \boldsymbol{\rho})$
2: Let $f(\cdot) := \text{-TRW}(\cdot; \vec{\boldsymbol{\theta}}, \boldsymbol{\rho})$; we use MFW to optimize over the contracted correction polytope $\text{conv}(V_\delta)$ where $V_\delta := (1 - \delta)V + \delta \boldsymbol{u}_0$.
3: Let $\epsilon$ be the desired accuracy of the approximate correction.
4: Let $\boldsymbol{\alpha}^{(0)}$ be such that $\boldsymbol{x}^{(0)} = \sum_{\boldsymbol{v} \in V_\delta} \alpha_{\boldsymbol{v}}^{(0)} \boldsymbol{v}$.
5: $\boldsymbol{x}^{(\text{new})} \leftarrow \textbf{MFW}(\boldsymbol{x}^{(0)}, \boldsymbol{\alpha}^{(0)}, V_\delta, \epsilon)$    *(see Algorithm 3)*
6: **return** $\boldsymbol{x}^{(\text{new})}$

---

**Algorithm 6:** Local Search using Iterated Conditional Modes, $f$ is the negative TRW objective

1: **LOCALSEARCH**$(\boldsymbol{x}^{(0)}, \boldsymbol{v}_{\text{init}}, \delta, \boldsymbol{\rho})$
2: $\boldsymbol{s}^{(0)} \leftarrow \boldsymbol{v}_{\text{init}}$
3: $V \leftarrow \emptyset$
4: **for** $k = 0 \dots \textbf{MAXITS}$ **do**
5:     $\tilde{\theta} = \nabla f(\boldsymbol{x}^{(k)}; \vec{\boldsymbol{\theta}}, \boldsymbol{\rho})$
6:     $\boldsymbol{s}^{(k+1)} \leftarrow \text{ICM}(-\tilde{\theta}, \boldsymbol{s}^{(k)})$    *(Approximate FW search using ICM;*
                        *we initialize ICM at previously found vertex $\boldsymbol{s}^{(k)}$)*
7:     $\boldsymbol{s}_{(\delta)}^{(k+1)} \leftarrow (1 - \delta)\boldsymbol{s}^{(k+1)} + \delta \boldsymbol{u}_0$
8:     $V \leftarrow V \cup \{\boldsymbol{s}^{(k+1)}\}$
9:     $\boldsymbol{d}_{(\delta)}^{(k)} \leftarrow \boldsymbol{s}_{(\delta)}^{(k+1)} - \boldsymbol{x}^{(k)}$
10:    Line-search: $\gamma_k \in \underset{\gamma \in [0,1]}{\arg \min} f\left(\boldsymbol{x}^{(k)} + \gamma \boldsymbol{d}_{(\delta)}^{(k)}\right)$
11:    Update $\boldsymbol{x}^{(k+1)} := \boldsymbol{x}^{(k)} + \gamma_k \boldsymbol{d}_{(\delta)}^{(k)}$    *(FW update)*
12: **end for**
13: **return** $\boldsymbol{x}^{(k+1)}, V$

---

# G  Comparison to perturbAndMAP

**Perturb & MAP.** We compared the performance between our method and perturb & MAP for inference on 10 node Synthetic cliques. We expand on the method we used to evaluate perturbAndMAP in Figure 1(e) and 1(f). We re-implemented the algorithm to estimate the partition function in Python (as described in Hazan and Jaakkola (2012), Section 4.1) and used toulbar2 (Allouche et al., 2010) to perform MAP inference over an inflated graph where every variable maps to five new variables. The log partition function is estimated as the mean energy of 10 exact MAP calls on the expanded graph where the single node potentials are perturbed by draws from the Gumbel distribution. To extract marginals, we fix the value of a variable to every assignment, estimate the log partition function of the conditioned graph and compute beliefs based on averaging the results of adding the unary potentials to the conditioned values of the log partition function.

# H  Correction Steps for Frank-Wolfe over $\mathcal{M}$

Recall that the correction step is done over the correction polytope, the set of all vertices of $\mathcal{M}$ encountered thus far in the algorithm. On experiments conducted over $\mathcal{M}$, we found that using a better correction algorithm often *hurt* performance. This potentially arises in other constrained optimization problems where the gradients are

Figure 4: Chinese Characters : Additional Experiments. TRBP (opt) denotes our implementation of tightening over $\mathbb{T}$ using a wrapper over libDAI (Mooij, 2010)

unbounded at the boundaries of the polytope. We found that better correction steps over the correction polytope (the convex hull of the vertices explored by the MAP solver, denoted $V$ in Algorithm 2), often resulted in a solution at or near a boundary of the marginal polytope (shared with the correction polytope). This resulted in the iterates becoming too small. We know that the Hessian of $\text{TRW}(\vec{\boldsymbol{\mu}}; \vec{\boldsymbol{\theta}}, \boldsymbol{\rho})$ is ill conditioned near the boundaries of the marginal polytope. Therefore, we hypothesize that this is because the gradient directions obtained when the iterates became too small are simply less informative. Consequently, the optimization over $\mathcal{M}$ suffered. We found that the duality gap over $\mathcal{M}$ would often increase after a correction step when this phenomenon occurred. The variant of our algorithm based on $\mathcal{M}_\delta$ is less sensitive to this issue since the restriction of the polytope bounds the smallest marginal and therefore also controls the quality of the gradients obtained.

# I  Additional Experiments

For experiments on the 10 node synthetic cliques, we can also track the average number of ILP calls required to converge to a fixed duality gap for any $\theta$. This is depicted in Figure 5(c). Optimizing over $\mathbb{T}$ realized three to four times as many MAP calls as the first iteration of inference.

Figure 4 depicts additional examples from the Chinese Characters test set. Here, we also visualize results from a wrapper around TRBPs implementation in libDAI (Mooij, 2010) that performs tightening over $\mathbb{T}$. Here too we find few gains over optimizing over $\mathbb{L}$.

Figure 5(a), 5(b) depicts the comparison of convergence of algorithm variants over $\mathcal{M}$ and $\mathcal{M}_\delta$ (same setup as Figure 1(a), 1(b)). Here, we plot $\zeta_\mu$.

(a) $\zeta_\mu$: $5 \times 5$ grid, $\mathcal{M}$ vs $\mathcal{M}_\delta$  (b) $\zeta_\mu$: 10 node clique, $\mathcal{M}$ vs $\mathcal{M}_\delta$  (c) Average ILP calls versus $\theta$: 10 node clique

Figure 5: Figure 5(a), 5(b) depict $\zeta_\mu$ corresponding to the experimental setup in Figure 1(a), 1(b) respectively. Figure 5(c) explores the average number of ILP calls taken to convergence with and without optimizing over $\boldsymbol{\rho}$

## J  Bounding $\log Z$ with Approximate MAP Solvers

Suppose that we use an approximate MAP solver for line 7 of Algorithm 2. We show in this section that if the solver returns an *upper bound* on the value of the MAP assignment (as do branch-and-cut solvers for integer linear programs), we can use this to get an upper bound on $\log Z$. For notational consistency, we consider using Algorithm 2 for $\min_{\boldsymbol{x} \in \mathcal{D}} f(\boldsymbol{x})$, where $f(\boldsymbol{x}) = -\text{TRW}(\vec{\boldsymbol{\mu}}; \vec{\boldsymbol{\theta}}, \boldsymbol{\rho})$ is convex, $\boldsymbol{x} = \vec{\boldsymbol{\mu}}$, and $\mathcal{D} = \mathcal{M}$.

The property that the duality gap may be used as a certificate of optimality (Jaggi, 2013) gives us:

$$f(\boldsymbol{x}^*) \geq f(\boldsymbol{x}^{(k)}) - g(\boldsymbol{x}^{(k)}) \implies -f(\boldsymbol{x}^*) \leq -f(\boldsymbol{x}^{(k)}) + g(\boldsymbol{x}^{(k)}). \tag{24}$$

Adding the gap onto the TRW objective yields an upper bound on the optimum (which from Equation 1 is an upper bound on $\log Z$), i.e. $\log Z \leq -f(\boldsymbol{x}^*)$. From our definition of the duality gap $g(\boldsymbol{x}^{(k)})$ (line 8 in Algorithm 2) and (24), we have:

$$\log Z \leq -f(\boldsymbol{x}^*) \leq -f(\boldsymbol{x}^{(k)}) + \left\langle -\nabla f(\boldsymbol{x}^{(k)}), \boldsymbol{s}^{(k)} - \boldsymbol{x}^{(k)} \right\rangle$$

$$= -f(\boldsymbol{x}^{(k)}) + \underbrace{\left\langle -\nabla f(\boldsymbol{x}^{(k)}), \boldsymbol{s}^{(k)} \right\rangle}_{\text{MAP call}} - \underbrace{\left\langle -\nabla f(\boldsymbol{x}^{(k)}), \boldsymbol{x}^{(k)} \right\rangle}_{\text{Can be computed efficiently}},$$

where $\boldsymbol{s}^{(k)} = \arg\min_{\boldsymbol{v} \in \mathcal{D}} \left\langle \nabla f(\boldsymbol{x}^{(k)}), \boldsymbol{v} \right\rangle = \arg\max_{\boldsymbol{v} \in \mathcal{D}} \left\langle -\nabla f(\boldsymbol{x}^{(k)}), \boldsymbol{v} \right\rangle$ (line 7 in Algorithm 2). Thus, if the approximate MAP solver returns an upper bound $\kappa$ such that $\max_{\boldsymbol{v} \in \mathcal{D}} \left\langle -\nabla f(\boldsymbol{x}^{(k)}), \boldsymbol{v} \right\rangle \leq \kappa$, then we get the following upper bound on the log-partition function:

$$\log Z \leq -f(\boldsymbol{x}^{(k)}) + \kappa - \left\langle -\nabla f(\boldsymbol{x}^{(k)}), \boldsymbol{x}^{(k)} \right\rangle. \tag{25}$$

For example, we could use a linear programming relaxation or a message-passing algorithm based on dual decomposition such as Sontag et al. (2008) to obtain the upper bound $\kappa$. There is a subtle but important point to note about this approach. Despite the fact that we may use a relaxation of $\mathcal{M}$ such as $\mathbb{L}$ or the cycle relaxation to compute the upper bound, we evaluate it at $\vec{\boldsymbol{\mu}}^{(k)}$ that is *guaranteed* to be within $\mathcal{M}$. This should be contrasted to instead optimizing over a relaxation such as $\mathbb{L}$ directly with Algorithm 2. In the latter setting, the moment we move towards a fractional vertex (in line 14) we would immediately take $\vec{\boldsymbol{\mu}}^{(k+1)}$ out of $\mathcal{M}$. Because of this difference, we expect that this approach will typically result in significantly tighter upper bounds on $\log Z$.

## Supplementary References

D. P. Bertsekas. *Nonlinear Programming*. Athena Scientific, Belmont, MA, 1999.

J. Guélat and P. Marcotte. Some comments on Wolfe's 'away step'. *Mathematical Programming*, 35(1):110–119, 1986.

Y. Nesterov. *Introductory Lectures on Convex Optimization*. Kluwer Academic Publishers, Norwell, MA, 2004.

C. Teo, A. Smola, S. Vishwanathan, and Q. Le. A scalable modular convex solver for regularized risk minimization. In *KDD*, 2007.

P. Wolfe. Convergence theory in nonlinear programming. In J. Abadie, editor, *Integer and Nonlinear Programming*, pages 1–23. North-Holland, 1970.