[Reviews · NeurIPS 2015]

Submitted by Assigned_Reviewer_1

This paper contains interesting technical details about how Frank-Wolfe-type algorithms can deal with the divergence of gradient at the boundary of the local/marginal polytope. This message however is somewhat muted in the way the claims were written.

For example the authors state that "convergent optimization TRW objective over the marginal polytope" is an open problem, and their claims focused on making the first such algorithm. I would argue that this claim matters only if one assumes the existence of a MAP oracle, otherwise one has a convergent algorithm that is outright infeasible -- all the combinatorial complexity is hidden in the NP-hard MAP solver.

I recommend that the authors limit the claims to improving FW-style algorithms. This way, the contributions are clear: how to guarantee convergence of FW (over both LOCAL and MARG), how to exploit an exact oracle MAP if one exists -- in that case we have a TRW optimizer over MARG, and if an approximate MAP exists, then we have an approximation of a TRW optimizer over MARG, that is different in nature than TRW optimizer over LOCAL -- although it is not clear how to understand exactly what happens when one uses approximate MAP (which is ok since empirically it seems to work).

The experiment section didn't highlight the cost vs benefit of doing this compared to another baseline. For example, how fast/slow this is compared to TRW-BP? (ok, I know that TRW-BP might have convergence issue, but it is informative to know convergence is achieved at what cost). It might be the case that FW might be slow (or even very slow) compare to message passing algorithm. Is that the case or not? stating this information is useful, even if FW is indeed slower.

As the approximate MAP solver, the authors run 3 different solvers and take the maximum (l 297). This is fine, but absolute CPU time matters here (a single MAP call turns into 3 actual calls).

Also convergence message passing algorithm for TRW over local relaxation was described in Globerson and Jaakkola 2007. How slow/fast this is compare to that approach?

How much time were spent in optimizing the edge appearance?

In general the result section were not clearly written.

Line 305: you said 15 trials with various width. Why only result for 5x5 is shown?

Why in Fig 1(b) error is going up?

What's the difference between the x-axis labels "updates to \rho" and "spanning trees iterations" in 2(a) and 2(b)?

Fig 2. label "L" isn't explained.

Captions for figure 1 and 2 are very hard to read.

Fig 2(c) and (d) were obtained on which model?

I can't decode footnote 2.

l 129. Please cite Sontag and Jaakola 2007 (who as far as I know were the first to propose the FW framework for TRW) in addition to Belanger 2013.
Summary: The paper proposed a method that uses FW to optimize the TRW objective. The idea of using FW for TRW optimization is a simple idea, is not new and goes back to Sontag and Jaakola 2007. What is new here is a convergent method by simply staying away from the boundary of the marginal (or local) polytope -- where the (TRW) entropy has infinite gradient by a certain distance --

and adapt this distance.

Submitted by Assigned_Reviewer_2

Overview: The present manuscript introduces an approach to approximate marginal inference in graphical models that is based on the Frank-Wolfe/conditional gradient method.

The main objective is to optimize over the full marginal polytope (a constraint set that grows exponentially) instead of the local polytope (a relaxation of the marginal polytope that typical message passing algorithms optimize over).

The presented approach works by repeatedly solving a linearization of the original problem, which in this specific case corresponds to MAP inference, with the natural/theta parameters determined by the current iterate.

A specific problem that should be addressed in this context is that the gradient is unbounded at the boundary of the marginal polytope. A barrier-like approach, inspired by interior point methods, is introduced to handle this situation and aid convergence. The empirical results demonstrate improvements versus optimization over the local polytope.

Positive points: + This paper provides interesting new perspectives on optimization-based/variational approximate marginal inference in graphical models.

While a lot of recent work has focused on "smoothing" the MAP problem to achieve faster convergence (thereby effectively marginalizing), in this work, the smooth marginalization problem is solved by a sequence of non-smooth MAP problems. This highlights again the close connection between the two problems (MAP/marginalization) and provides interesting perspectives. + Related work is discussed in sufficient detail, and the work is well embedded into its scientific context. + The theory regarding the "barrier" approach is developed in great detail in the supplementary material.

It is clear that a lot of effort went into development of the approach. + The material developed in the paper seems to be technically correct.

Negative points: - I have some criticism regarding the approach in general (even though I find it interesting, from a pure curiosity-based research perspective).

The use of the Frank-Wolfe method is indicated if optimization over a *non-linear* objective function is not plausible due to the difficulty of the constraint set, but optimization of a linear objective function over the same constraint set can be handled gracefully.

However, this is not the case here.

Optimization of a linear objective function over the exact marginal polytope is NP-hard as well.

How much is really gained here?

In fact, one could argue that it is even harder to solve the MAP problem, sine it is non-smooth. Many recent approaches have focused on smoothing the MAP problem, effectively turning it into a smooth marginalization problem, precisely to improve convergence.

What must be done in practice, with the proposed approach, is to solve the inner MAP problem approximately, thereby voiding guarantees.

One might argue that it is more elegant to establish a well-defined approximation in the first place, and to then solve the approximation exactly.

- The manuscript is purely concerned about approximation quality, and not at all about computational cost. As an example, when evaluating the impact of using approximate MAP solutions in the inner loop, the authors run three solvers (QPBO, TRW-S, ICM) and use the result that realizes the highest energy.

Clearly, this is not very practical outside of synthetic experiments.

- In the discussion (section 6), the authors refer to their approach as performing "exact marginal inference".

This is not the case: While the constraint set is exact, the objective function still uses an approximation of the exact entropy, the TRW bound.

I also do not find it true that tightening over the spanning tree polytope has remained relatively unexplored in recent work.

This was done both in the original work of Wainwright et al. (2005), as well as the more recent work of Jancsary and Matz (2011).

- Figures 1 and 2 requires further polishing: The caption lines run over the space allocated to each specific sub-figure, making the captions very hard to read.

In Figure 2 (a), the label of the Y coordinate states "Error in Marginals", but the caption states "Error in logZ".

In general, the experiments section and the figures therein leave a rather disorganized impression. I assume that this is partially due to the lack of space, but it seems that further improvements could be achieved here.

- A question regarding Figures 2 (c) and (d), and the statement in Section 5.1: How can the optimization of the edge occurrence probabilities rho possibly not have any effect on the tightness of the bound on on logZ if optimizing over the local polytope? Rho is expected to have a significant influence on the quality of the entropy approximation, both when optimizing over the marginal polytope *and* when optimizing over the local polytope.

Both in Wainwright et al. (2005) and Jancsary and Matz (2011), the improvement of the bound was considerable in simulations.

Is this an artifact of the specific type of interactions chosen, or does it perhaps point to a problem in the code/experimental setup?

- A question regarding Figure 3: I assume that for TRBP, no optimization over rho was performed (this relates to my previous question - why should optimization over rho not help)?

This is a different problem now than in Figure 2, so even if optimization over rho did not help there, one cannot in general conclude that it would not help for the Chinese characters problem.

It seems somewhat unfair to compare the proposed method *with* optimization over rho to TRBP *without* such optimization.

- Minor point: Footnote 2 is a fragment -

"which converges which iterates differing by 0.002"

Summary: This manuscript provides interesting new perspectives on approximate marginal inference in graphical models.

However, the manuscript would benefit from some more polishing, and the experimental evaluation raises some questions.

Submitted by Assigned_Reviewer_3

The submitted paper considers marginal inference using the TRW objective. The authors introduce an algorithm that is globally convergent therefore and internally uses the Frank-Wolfe algorithm. In experiments, the authors provide various compelling results for synthetic and real-world instances.

Quality. High. Well written and interesting paper.

Clarity. The main paper is pretty dense, mainly due to the lack of space. However, the authors provide an extensive supplementary that contains all proofs and the further experiments. It remains unclear, what effect approximate MAP inference has.

Originality. The paper improves over existing result by introducing the first globally convergent algorithm for marginal inference using the TRW objective (I didn't check all math in detail though).

Significance. I expect this paper to inspire multiple followup works. Furthermore, the provided algorithms can be applied in various applications and potentially improve results in the corresponding areas. The presented ideas are not limited to the TRW objective.
Summary: The paper is well written, contains several interesting ideas that can be applied in various fields, and the presented algorithm tackles an interesting open problem in variational inference. The paper could span interesting followup works.

Submitted by Assigned_Reviewer_4

This work proposes an algorithm for marginal inference in graphical models, which optimises

the TRW objective on the marginal polytope based on Frank-Wolfe algorithm. It follows the direction of using MAP inference to do the marginal inference. To reduce the number of MAP inference, it introduces several heuristic: namely correction step which re-optimise over the convex hull of points encountered, and contraction of the polytope to avoid regions near the boundary. It also derives a convergence bound on the sub-optimality of the proposed algorithm.

* This paper is generally clear but requires further clarifications in several points.

* Using correction steps is one of the major contribution of this paper. Thus, I suggest the authors provide a more precise algorithmic definition of the correction step. From the description in line 137-139 or 268 - 270, it is hard for a reader to implement the algorithm and reproduce the results.

* When introducing correction step and contraction of marginal polytope, it would be good to provide references on similar ideas to help understand the background of these two ideas.

* Theorem 3.6:

- Does this result only hold for exact MAP inference? Will approximate inference affect the result?

- Does this result also hold for Frank-Wolfe based

TRW objective optimisation without correction step and contraction of marginal polytope? If not, please provide analysis on the reason.

* Experiments:

- On the synthetic dataset, it would be helpful to also include the comparison to TRBP. Also, please be clear on which MAP inference algorithm, among those mentioned in line 297-301 is used on each experiment setting.

- For experiments on Horses and Chinese Characters, quantitative comparisons should also be included. It is hard to draw conclusions from the example output comparison only.

Summary: The approach proposed in this paper is well motivated and interesting but the writing could be further improved.

Author Feedback
Author rebuttal: We thank the reviewers for their insightful reviews and feedback.

A] Practicality and reliance on MAP inference [R1, R5, R7]
Our algorithm is most useful for settings where the local relaxation L is loose, but where MAP inference is nonetheless not so difficult. An example is the Chinese character dataset. Recent work on fast approximate MAP solvers have been shown to be very successful in these settings, e.g. those that solve the cycle relaxation (Sontag et al., 2007), solvers in computer vision such as QPBO, and solvers for RBMs such as Wang et al '13. Importantly, any approximate MAP solver that provides an upper bound on MAP can be used in conjunction with our algorithm to provide upper bounds on logZ (we will edit the algorithm boxes and description to make this clear). When these bounds are tighter than L, our algorithm can be expected to give more accurate marginals and tighter bounds on logZ than TRW.

B] Approximate MAP solvers [R2, R3]
For Thms 3.6 and 3.7, using standard techniques one could get the same convergence rate assuming a multiplicative guarantee on the gap on step 8 of Alg. 2 with an approx. MAP oracle (this only changes a multiplicative constant in the rate), as was done for FW-like algorithms (see e.g. Theorem C.1 in Lacoste-Julien et al. ICML 2013). Similarly, with an eps-additive error guarantee instead, we can also prove the same convergence rates, but up to a suboptimality error of eps.

C] Relevance of Contraction and Correction Steps [R3, R4]
Thms 3.6 and 3.7 apply with or without the correction step (pseudocode given in Alg. 5 of the supplementary). However, without contraction we do not believe it is possible to prove a convergence rate for vanilla FW with line-search, because iterates can get arbitrarily close to the boundary of the polytope where the Hessian is unbounded.

D] Experimental Section [R1, R5, R7]
Multiple (approximate) MAP calls per iteration can be easily incorporated in a multi-core environment by parallelizing them. E.g., the state-of-the-art ILP solver Gurobi uses concurrent optimization by default. We will revise the figures and captions.

[R1]
Thanks for pointing out the interesting connections to smoothing the MAP problem!

"Rho is expected to have a significant influence.... problem in the code/experimental setup?"
For theta = 0.5 (see Fig. 2c), we see an improvement (~0.01) in the error in marginals for L vs L(rho opt). Wainwright et al. '05 have experiments based on potentials sampled in the range [0, 0.5], whereas ours span [0.5, 8] which represent significantly stronger interactions. In their JMLR paper, Figure 10(b) does not indicate much gain from optimizing rho for cliques with theta = 0.5. Jancsary and Matz '11 sample potentials from N(0,1) and also find differences in marginal error on the order of ~0.01 (c.f. Table 1 Column: Complete ExpGauss). Regarding Figure 3 (Chinese characters), we experimented with optimizing over rho using a wrapper around TRBP, but found no significant differences in the results.

[R3]
"For experiments on Horses and Chinese Characters, quantitative comparisons should also be included"
Exact inference is infeasible on problems of these size. What we can compare in the next revision is upper bounds on logZ. We expect the quantitative results to mirror the qualitative ones, i.e. similar for Horses and tighter bounds for the Chinese characters.

[R5]
"convergence rate is sub-linear"
Given that the TRW objective has non-Lipschitz gradients and that the set being optimized over has no fast proximal oracle, we would not be surprised if a sub-linear rate of convergence is the best that one can achieve (it is an open question). The heuristics we adopt aid convergence.

"good initialization would be important in practice."
In practice, within the first few iterations of FW, the algorithm finds a good set of extrema of the marginal polytope and a correction step places the iterate close to the optimal solution; therefore, it's unclear how much initialization matters.

[R7]
"experiment section didn't highlight the cost vs benefit"
In an optimized implementation, we expect one approx. MAP call to take as long as a complete solve of TRW over L (using the better of TRBP, Globerson/Jaakkola, or Jancsary/Matz). For a rough estimate, on the 5x5 Grids Figure 1(a),(b), we find that with runtime equivalent to running TRBP 5 times, our estimate of logZ is better than that over L. The marginals we obtain are immediately better.

"authors limit claims to improving FW-style algorithms. This way, the contributions are clear"
We thank the reviewer for this suggestion and will reframe the introduction based on this comment.

We updated edge appearances 10 times. The error in 1(b) is going up because the entropy approximation from the initial setting of rho is poor. This is an interesting phenomenon that is only apparent when using a FW algorithm. Fig. 2(c)(d) correspond to 10 node synthetic cliques.